# Spider venom-derived peptide induces hyperalgesia in Na$_v$1.7 knockout mice by activating Na$_v$1.9 channels

Xi Zhou [1,5], Tingbin Ma[2,5], Luyao Yang [2], Shuijiao Peng[1], Lulu Li[2], Zhouquan Wang[1], Zhen Xiao[1], Qingfeng Zhang[1], Li Wang[1], Yazhou Huang[1], Minzhi Chen[1], Songping Liang[1], Xianwei Zhang[3], Jing Yu Liu [2,4✉] & Zhonghua Liu [1✉]

The sodium channels Na$_v$1.7, Na$_v$1.8 and Na$_v$1.9 are critical for pain perception in peripheral nociceptors. Loss of function of Na$_v$1.7 leads to congenital insensitivity to pain in humans. Here we show that the spider peptide toxin called HpTx1, first identified as an inhibitor of K$_v$4.2, restores nociception in Na$_v$1.7 knockout (Na$_v$1.7-KO) mice by enhancing the excitability of dorsal root ganglion neurons. HpTx1 inhibits Na$_v$1.7 and activates Na$_v$1.9 but does not affect Na$_v$1.8. This toxin produces pain in wild-type (WT) and Na$_v$1.7-KO mice, and attenuates nociception in Na$_v$1.9-KO mice, but has no effect in Na$_v$1.8-KO mice. These data indicate that HpTx1-induced hypersensitivity is mediated by Na$_v$1.9 activation and offers pharmacological insight into the relationship of the three Na$_v$ channels in pain signalling.

[1] The National and Local Joint Engineering Laboratory of Animal Peptide Drug Development, College of Life Sciences, Hunan Normal University, Changsha 410081, China. [2] Key Laboratory of Molecular Biophysics of the Ministry of Education, College of Life Science and Technology, Huazhong University of Science and Technology (HUST), Wuhan 430074, China. [3] Department of Anesthesiology, Tongji Hospital of HUST, Wuhan 430030, China. [4] Institute of Neuroscience, State Key Laboratory of Neuroscience, CAS Center for Excellence in Brain Science and Intelligence Technology, Chinese Academy of Sciences, Shanghai 200031, China. [5] These authors contributed equally: Xi Zhou, Tingbin Ma. ✉email: liujy@ion.ac.cn; liuzh@hunnu.edu.cn

Pain is an unpleasant sensory and emotional experience associated with actual or potential tissue damage, and is a serious public health issue[1]. However, pain is also a part of the body's warning mechanism, cautioning humans to take action to prevent further tissue damage[2–4]. The voltage-gated sodium channel ($Na_v$) $Na_v1.7$ is known to play critical roles in the regulation of peripheral pain. Homozygous or compound heterozygous loss-of-function mutations in $Na_v1.7$ lead to congenital insensitivity to pain (CIP)[2,5,6], whereas gain-of-function mutations cause episodic pain (i.e., primary erythromelalgia and paroxysmal extreme pain disorder) in humans[7–9]. This evidence indicates that selectively blocking $Na_v1.7$ may be useful to relieve pain. Sustained efforts have been made to develop selective inhibitors of this channel, some of which have shown efficacy in clinical studies, although larger clinical trials are needed to definitively assess efficacy[10–13]. On the other hand, compared with painful, painless may be more serious. CIP individuals cannot distinguish between sharp and dull stimuli, leading to self-mutilation and painless fractures, and some individuals are even unable to detect temperature differences[2,5,6]. Therefore, therapeutics that restore the pain responses in $Na_v1.7$-related CIP would be useful. Notably, $Na_v1.7$ might not be a suitable direct target to achieve such an effect because of its loss of function in affected individuals; therefore, an alternative strategy should be developed. $Na_v1.7$ and two other $Na_v$ subtypes, $Na_v1.8$ and $Na_v1.9$, are preferentially expressed in the peripheral terminals of sensory neurons[14]. More recently, genetic and functional studies have illustrated that mutations in $Na_v1.8$ and $Na_v1.9$ cause human pain disorders, providing direct clinical evidence linking these two channels to human pain[3,15–17]. Considering that the three channels play distinct roles in the generation and propagation of action potentials (APs)[18–20], they might regulate pain signaling cooperatively, and the elucidation of their relationship in pain regulation may be helpful for the treatment of $Na_v1.7$-related CIP.

Animal venom is a rich source of $Na_v$ modulators and potential therapeutic compounds[21–26]. Specific modulators are not only novel drug candidates for therapeutics but also powerful pharmacological tools to probe the physiological roles of $Na_v$s[23,25–28]. In this study, we screened pain-inducing compounds from animal venoms and discovered that a spider peptide toxin, HpTx1, was able to rescue the pain response in $Na_v1.7$ knockout ($Na_v1.7$-KO) mice by activating $Na_v1.9$. HpTx1 was previously identified as an inhibitor of the voltage-gated potassium channel ($K_v$) $K_v4.2$[29], but our study showed that it is also a $Na_v$ modulator. Here, we provide pharmacological evidence that cross talk among $Na_v1.7$, $Na_v1.8$, and $Na_v1.9$ may affect AP firing and pain signaling, thereby establishing an important role for $Na_v1.9$ in pain perception.

## Results

**HpTx1 induces pain responses in $Na_v1.7$-KO mice.** To discover peptide activators that can induce pain and recover pain responses in $Na_v1.7$-related CIP, we first fractionated 15 crude venoms (from ten spiders and five snakes) using semipreparative reversed-phase high-performance liquid chromatography (RP-HPLC) and collected a total of 110 fractions (5–10 fractions per venom). Six of these fractions were identified to have pain-inducing activity, with a fraction from the venom of the spider *H. venatoria* exhibiting the strongest efficacy (Fig. 1a). By further purifying this fraction by analytical RP-HPLC, we identified a component with such a pain-inducing efficacy and named it HpTx1 (rational nomenclature: κ-sparatoxin-Hv1a). The molecular weight of HpTx1 was 3910.8 Da as determined by matrix-assisted laser desorption/ionization-time of-flight mass

spectrometry (MALDI–TOF MS) (Supplementary Fig. 1a). HpTx1 was found to have some sequence similarity, especially a conserved cysteine pattern, with some spider peptide toxins adopting an inhibitor cystine knot (ICK) motif (Fig. 1b), suggesting that the space structure of HpTx1 might contain a typical ICK motif.

We next examined whether HpTx1 could produce pain sensation in mice lacking $Na_v1.7$. According to methods reported previously[30,31], we generated $Na_v1.7$-KO mice by crossing $Na_v1.7$ floxed mice (f$Na_v1.7$ in C57BL/6 genetic background) with $Na_v1.8$-Cre mice, which led to the specific deletion of $Na_v1.7$ in $Na_v1.8$-positive sensory neurons (tissue-restricted $Na_v1.7$ knockout, $Na_v1.7$-KO). These mice showed deficits in mechanical pain responses but no alteration in thermal pain behavior (Fig. 1d, e), consistent with the results reported by Minett et al.[31]. As shown in Fig. 1c, injection of 10 μM HpTx1 into the hind paws of $Na_v1.7$-KO mice or control (f$Na_v1.7$) littermate mice triggered robust nocifensive behaviors, such as licking and biting of the injected paws. Furthermore, the pain-inducing effect was further validated in evoked pain models (Fig. 1d, e). $Na_v1.7$-KO mice treated with 10 μM HpTx1 recovered the deficit in mechanical pain caused by $Na_v1.7$ ablation (Fig. 1d); injection of 10 μM HpTx1 also reduced thresholds for thermal pain in $Na_v1.7$-KO mice, paralleling the effect of HpTx1 on f$Na_v1.7$ mice (Fig. 1e).

Unlike intraplantar injection of 10% formalin, which elicited robust neurogenic inflammation in the injected hind paws, HpTx1 injection failed to produce neurogenic inflammation, as revealed by the Evans blue test (Fig. 1f, g). No swelling was observed in hind paws injected with 50 μM HpTx1, whereas serious edema was found with injection of 10% formalin (Fig. 1h). These results suggested that HpTx1 evoked pain behaviors in f$Na_v1.7$ and $Na_v1.7$-KO mice, but failed to trigger neurogenic inflammation.

**HpTx1 activates some small DRG neurons in $Na_v1.7$-KO mice.** The effects of HpTx1 on membrane excitability were examined in small (<30 μm) dorsal root ganglion (DRG) neurons from WT mice by using current-clamp recordings. Four parameters related to AP firing, including resting membrane potential (RMP), current threshold (rheobase), amplitude, and firing frequency, were determined in the experiments. As shown in Fig. 2a, 0.75 μM HpTx1 significantly depolarized RMP by ~2.0 mV (control: $-50.4 \pm 1.2$ mV; HpTx1: $-48.4 \pm 1.3$ mV; $n = 30$, $P < 0.0001$) (Supplementary Table 1). Treatment with 0.75 μM HpTx1 remarkably decreased the rheobase to evoke an AP by 9.3 pA (control: $41.3 \pm 3.9$ pA; HpTx1: $32.0 \pm 3.4$ pA; $n = 30$, $P = 0.0026$) (Fig. 2b; Supplementary Table 1). Importantly, 15 out of the 30 DRG neurons tested (50.0%) exhibited a decrease in rheobase in the presence of HpTx1, whereas 10% exhibited an increase, and 40% showed no change (Supplementary Table 2). However, no significant changes in AP amplitude were observed in the presence of 0.75 μM HpTx1 (control: $117.8 \pm 1.3$ mV; HpTx1: $116.8 \pm 1.4$ mV; $n = 30$, $P = 0.1$) (Fig. 2c; Supplementary Table 1). In 10 out of the 15 neurons with reduced rheobase, the reduction in rheobase was also associated with a prominent increase in firing frequency in response to depolarizing currents (Fig. 2d). The input resistance remained unchanged by HpTx1 treatment (Supplementary Table 1). Similar effects of HpTx1 on membrane excitability were observed in some small DRG neurons from f$Na_v1.7$ mice (Fig. 2a–c; Supplementary Table 1).

The effects of HpTx1 on membrane excitability were also assessed in small DRG neurons from $Na_v1.7$-KO mice. As shown in Fig. 2e, 0.75 μM HpTx1 significantly depolarized RMP by 3.2 mV (control: $-49.6 \pm 1.0$ mV; HpTx1: $-46.4 \pm 1.1$ mV; $n = 28$, $P = 0.006$) and decreased the rheobase by 16.6 pA (control:

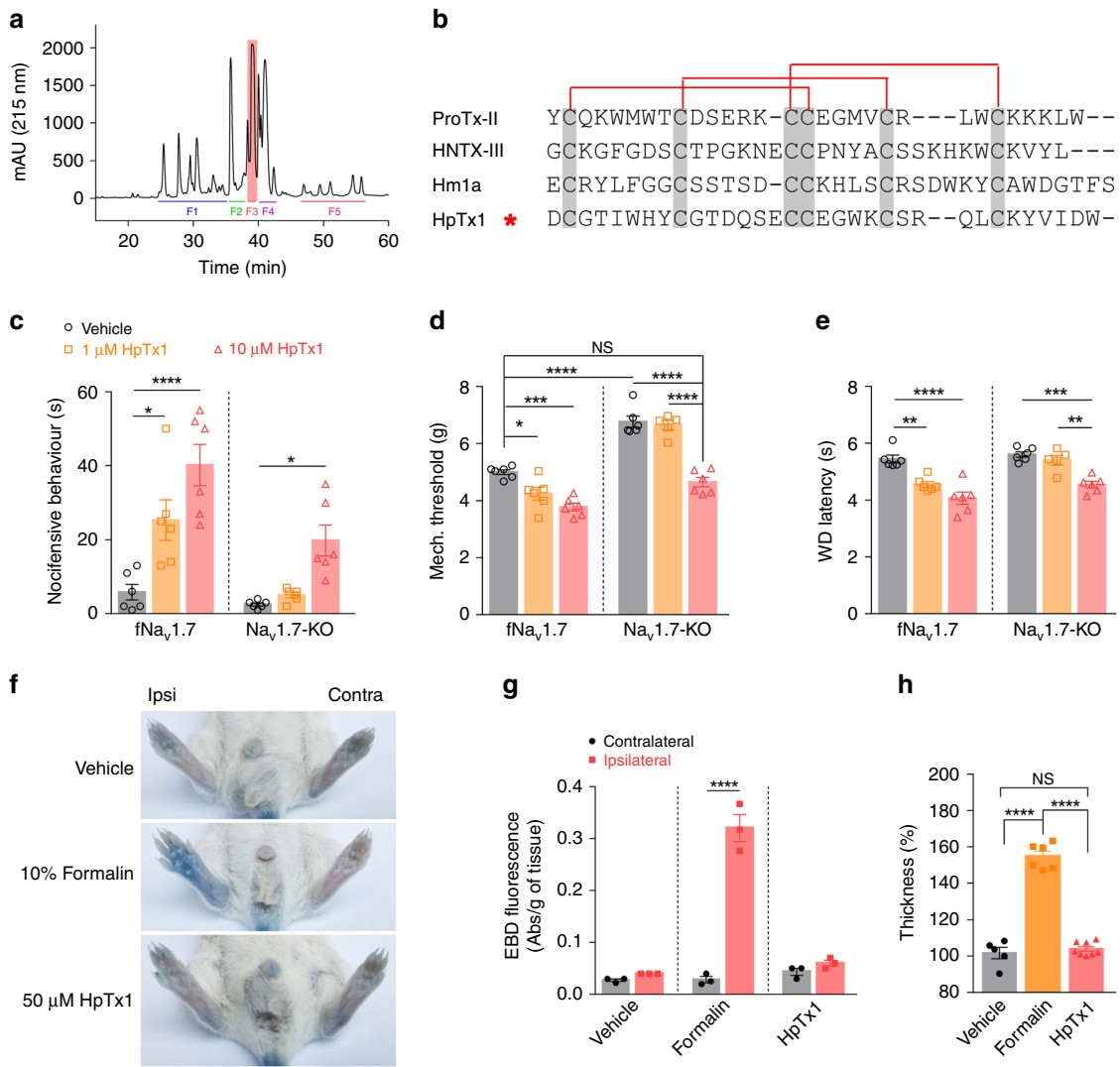

**Fig. 1 HpTx1 rescues the pain response in Na$_v$1.7-KO mice. a** RP-HPLC profile of the venom from the spider *H. venatoria*. The F3 fraction contains HpTx1 (pink). **b** Sequence alignment of HpTx1 with several ICK toxins; red lines show the disulfide linkage. **c** Comparison of nocifensive behaviors (licking or biting) following intraplantar injection of vehicle (10 µl 0.9% saline, $n = 6$) versus HpTx1 (1 µM or 10 µM in 10 µl saline, $n = 6$) (two-way ANOVA followed by Tukey's multiple comparisons test, treatment × genotype: $F_{(2,30)} = 3.447$, $P = 0.0449$; treatment: $F_{(2,30)} = 24.05$, $P < 0.0001$; genotype: $F_{(1,30)} = 23.09$, $P < 0.0001$). **d** Mechanical response thresholds measured in paws in response to vehicle (black circles, $n = 6$), 1 µM HpTx1 (yellow squares, $n = 6$ for fNa$_v$1.7 mice; $n = 5$ for Na$_v$1.7-KO mice) or 10 µM HpTx1 (red triangles, $n = 6$) injections (two-way ANOVA followed by Tukey's multiple comparisons test, treatment × genotype: $F_{(2,29)} = 10.52$, $P = 0.0004$; treatment: $F_{(2,29)} = 54.72$, $P < 0.0001$; genotype: $F_{(1,29)} = 150.2$, $P < 0.0001$). **e** Latency of paw withdrawal (WD) to a noxious thermal stimulus measured after intraplantar injection of vehicle (black circles, $n = 6$), 1 µM HpTx1 (yellow squares, $n = 6$ for fNa$_v$1.7 mice; $n = 5$ for Na$_v$1.7-KO mice) or 10 µM HpTx1 (red triangles, $n = 6$) (two-way ANOVA followed by Tukey's multiple comparisons test, treatment × genotype: $F_{(2,29)} = 2.857$, $P = 0.0737$; treatment: $F_{(2,29)} = 36.83$, $P < 0.0001$; genotype: $F_{(1,29)} = 17.71$, $P = 0.0002$). **f** Images of hind paws with Evans blue staining. Ipsi ipsilateral paws, Contra contralateral paws. **g** Quantification of Evans blue staining in ipsilateral and contralateral hind paws ($n = 3$, two-way ANOVA followed by Tukey's multiple comparisons test, treatment × paw: $F_{(2,12)} = 96.1$, $P < 0.0001$; treatment: $F_{(2,12)} = 87.79$, $P < 0.0001$; paw: $F_{(1,12)} = 128.9$, $P < 0.0001$). **h** Relative thickness of injected hind paws normalized to that of uninjected ones ($n = 5$ for vehicle, $n = 6$ for formalin, $n = 8$ for HpTx1, one-way ANOVA followed by Tukey's multiple comparisons test: $F_{(2,16)} = 160.9$, $P < 0.0001$). Data are represent the mean ± S.E.M. $^*P < 0.05$, $^{**}P < 0.01$, $^{***}P < 0.001$, $^{****}P < 0.0001$. Exact $P$ (**c–e**, **g**, **h**) are presented in Supplementary Data 1. Source data are provided as a Source Data file.Source Data file.

48.3 ± 4.6 pA; HpTx1: 31.7 ± 3.4 pA; $n = 29$, $P = 0.0003$), but did not alter AP amplitude (control: 108.2 ± 1.3 mV; 0.75 µM HpTx1: 106.5 ± 1.3 mV; $n = 29$, $P = 0.09$) (Supplementary Table 1) or input resistance (Supplementary Table 1). A total of 65.5% of neurons (19 out of 29) showed a decrease in rheobase in the presence of HpTx1 (Supplementary Table 2), and these neurons that exhibited a reduction in rheobase also showed enhanced AP firing in response to depolarizing currents (Fig. 2f, g). These results indicated that HpTx1 might enhance the membrane

excitability of some small DRG neurons from WT, fNa$_v$1.7, and Na$_v$1.7-KO mice by depolarizing RMP, decreasing current threshold, and increasing AP firing.

**HpTx1 is an inhibitor of Na$_v$1.7 and an activator of Na$_v$1.9.** Sanguinetti et al.[29] demonstrated that HpTx1 is a blocker of the K$_v$4.2 channel. Our study confirmed HpTx1 inhibition of K$_v$4.2 currents, with a half-maximal inhibitor concentration (IC$_{50}$) value measured at 1.2 ± 0.3 µM (Supplementary Fig. 1b). K$_v$4.2 is

 3

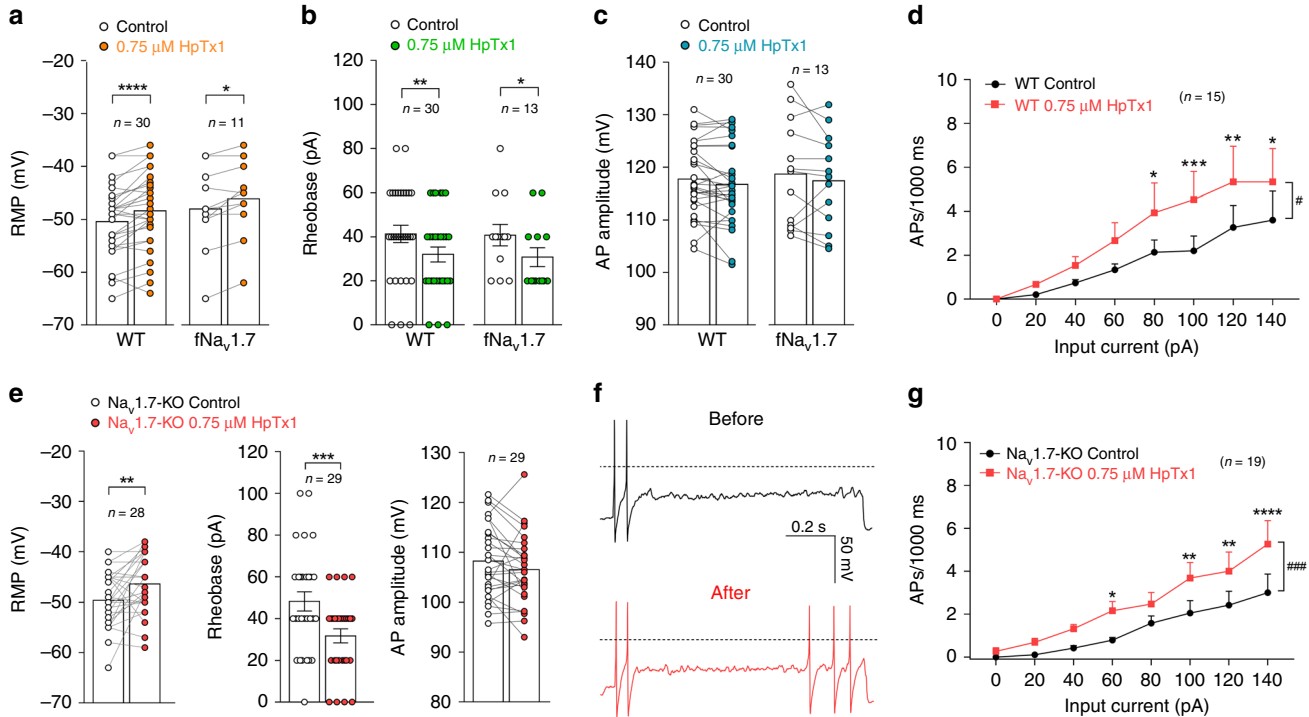

**Fig. 2 HpTx1 activates some small DRG neurons in WT and Na$_v$1.7-KO mice. a–d** Current-clamp recording shows that HpTx1 enhances the excitability of small (<30 μm) DRG neurons from WT and fNa$_v$1.7 mice. Bars show significant changes for RMP (**a**, two-way repeated measures ANOVA followed by Bonferroni's multiple comparisons test, treatment × genotype: $F_{(1,39)} = 0.02197$, $P = 0.8829$; treatment: $F_{(1,39)} = 22.12$, $P < 0.0001$; genotype: $F_{(1,39)} = 0.9114$, $P = 0.3456$) and rheobase (**b**, two-way repeated measures ANOVA followed by Bonferroni's multiple comparisons test, treatment × genotype: $F_{(1,41)} = 0.01839$, $P = 0.8928$; treatment: $F_{(1,41)} = 15.47$, $P = 0.0003$; genotype: $F_{(1,41)} = 0.02329$, $P = 0.8795$), but no effect on AP amplitude (**c**) in the presence of 0.75 μM HpTx1. **d** Statistics plots show significant increases in AP spike number in the presence of 0.75 μM HpTx1 ($n = 15$, two-way repeated measures ANOVA followed by Bonferroni's multiple comparisons test, treatment × inject current: $F_{(7,98)} = 2.228$, $P = 0.0382$; treatment: $F_{(1,14)} = 7.716$, #$P = 0.0148$; inject current: $F_{(7,98)} = 8.916$, $P < 0.0001$). **e–g** Current-clamp recordings show that HpTx1 increases the excitability of small DRG neurons from Na$_v$1.7-KO mice. **e** Bars show significant changes for RMP (left, $n = 28$, parametric paired two-tailed t test: $t_{27} = 3.0$, $P = 0.006$) and rheobase (middle, $n = 29$, nonparametric Wilcoxon matched-pairs signed-rank test: $P = 0.0003$), but no effect for AP amplitude (right, $n = 29$, parametric paired two-tailed t test: $t_{28} = 1.7$, $P = 0.093$) in the presence of 0.75 μM HpTx1. **f** AP traces recorded from a representative Na$_v$1.7-KO mouse DRG neuron before (black) and after (red) the application of HpTx1. The dashed lines indicate 0 mV. **g** Statistics plots show significant increases in AP spike number in the presence of 0.75 μM HpTx1 ($n = 19$, two-way repeated measures ANOVA followed by Bonferroni's multiple comparisons test, treatment × inject current: $F_{(7,126)} = 2.313$, $P = 0.0298$; treatment: $F_{(1,18)} = 17.69$, ###$P = 0.0005$; inject current: $F_{(7,126)} = 14.2$, $P < 0.0001$). All DRG neurons recorded were held at $-53 \pm 2$ mV. Error bars represent the mean ± S.E.M. *$P < 0.05$, **$P < 0.01$, ***$P < 0.001$, ****$P < 0.0001$. Exact $P$ (**a**, **b**, **d**, **g**) are presented in Supplementary Data 1. Source data are provided as a Source Data file.Source Data file.

crucial for the ERK-dependent modulation of dorsal horn neuronal excitability and the ERK-dependent hyperalgesia, but it is seldomly expressed in peripheral sensory neurons[32,33]. We further found that HpTx1 had no evident effect on voltage-gated potassium channels in DRG neurons, and the current–voltage curves were not altered by HpTx1 treatment (Supplementary Fig. 1c, d). Therefore, we assumed that potassium channel inhibition might not be the primary reason for pain reactions induced by HpTx1. Moreover, HpTx1 did not change voltage-gated calcium channels in DRG neurons (Supplementary Fig. 1e). Therefore, we investigated the effect of HpTx1 on Na$_v$s. HpTx1-induced pain responses were not distinct between the WT and Na$_v$1.7-KO mice, suggesting that the responses are independent of Na$_v$1.7. Unexpectedly, HpTx1 inhibited the human Na$_v$1.7 (hNa$_v$1.7) currents expressed in human embryonic kidney (HEK) 293T cells with an IC$_{50}$ value of $0.51 \pm 0.12$ μM (Fig. 3a) without altering the steady-state activation (control: V$_{1/2} = -21.9 \pm 1.9$ mV and k $= 5.2 \pm 0.2$ mV; HpTx1: V$_{1/2} = -22.1 \pm 3.3$ mV and k $= 6.9 \pm 0.3$ mV; $n = 4$, $P > 0.05$) or inactivation (control: V$_{1/2} = -70.7 \pm 1.9$ mV and k $= -4.8 \pm 0.1$ mV; HpTx1: V$_{1/2} = -73.5 \pm 1.7$ mV and k $= -4.7 \pm 0.1$ mV; $n = 7$, $P > 0.05$) of this channel (Fig. 3b; Supplementary Table 3). In addition,

examination of the effect of HpTx1 on tetrodotoxin-sensitive (TTX-S) Na$_v$ channels in WT mouse small DRG neurons demonstrated that 1 μM HpTx1 inhibited TTX-S Na$_v$ currents by $62.9 \pm 7.8\%$ (Fig. 3c). In addition to Na$_v$1.7, the TTX-S channel Na$_v$1.6 is also found within small DRG neurons. Indeed, HpTx1 had inhibitory activity on Na$_v$1.6 with an IC$_{50}$ of $5.63 \pm 0.13$ μM, indicating that HpTx1 was tenfold less potent at Na$_v$1.6 than at Na$_v$1.7 (Supplementary Fig. 1f). Vasylyev et al.[34] estimated that ~70% TTX-S Na$_v$ currents are mediated by Na$_v$1.7 in mouse small DRG neurons. These data suggested that the observed effect on TTX-S Na$_v$ currents mainly arise from Na$_v$1.7 inhibition by HpTx1.

From our understanding, inhibition of Na$_v$1.7 may produce analgesia rather than pain. This seemingly paradoxical result suggests that other ion channels might be involved in HpTx1-induced pain. Therefore, the two TTX-resistant (TTX-R) channels, Na$_v$1.8 and Na$_v$1.9, were examined. As shown in Fig. 3d, e and Supplementary Fig. 2a, HpTx1 robustly enhanced the current amplitude and inhibited the fast inactivation of TTX-R Na$_v$ channels in mouse small DRG neurons. Specifically, steady-state inactivation (SSI) was not significantly altered (Supplementary Fig. 2b), but a noninactivated component (~35–42% of the transient inward peak currents) was observed

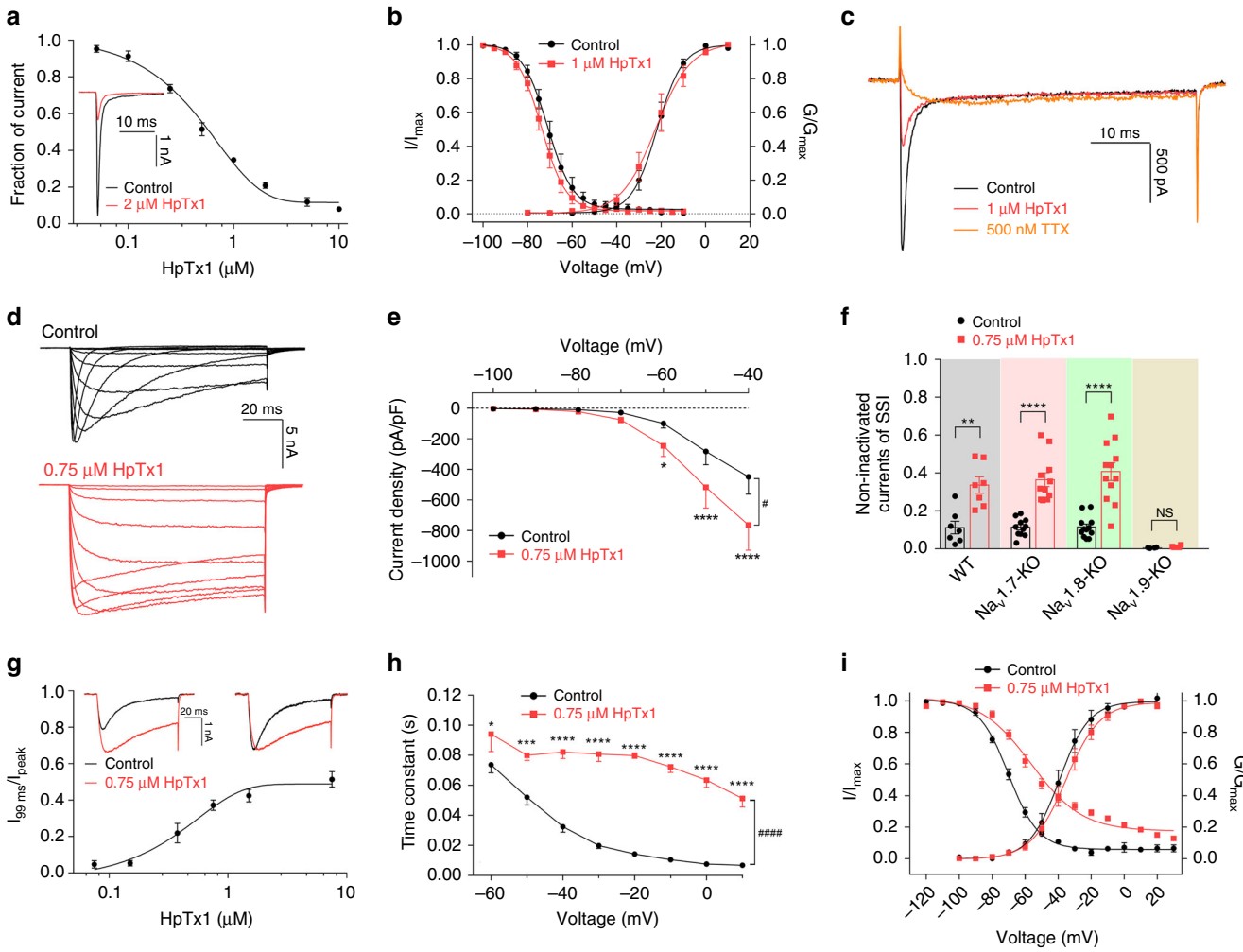

**Fig. 3 HpTx1 inhibits Na$_v$1.7 currents and enhances Na$_v$1.9 activity. a** The dose-dependent inhibition of hNa$_v$1.7 currents by HpTx1 ($n = 7$). The inset shows representative current traces in the presence (red) or absence (black) of 2 μM HpTx1. **b** Voltage-dependent steady-state activation (G/G$_{max}$, $n = 4$) and fast inactivation (I/I$_{max}$, $n = 7$) of hNa$_v$1.7 are not altered by 1 μM HpTx1. **c** Representative currents show the effect of 1 μM HpTx1 on TTX-S Na$_v$s in mouse small DRG neurons. **d, e** HpTx1 increases TTX-R Na$_v$ currents in mouse small DRG neurons and inhibits their fast inactivation, as shown by representative current traces (**d**) and current density (**e**, $n = 6$, two-way repeated measures ANOVA followed by Bonferroni's multiple comparisons test, treatment × voltage: $F_{(6,30)} = 9.099$, $P < 0.0001$; treatment: $F_{(1,5)} = 15.41$, #$P = 0.0111$; voltage: $F_{(6,30)} = 16.25$, $P < 0.0001$). **f** Bars show noninactivated components observed in the steady-state inactivation (SSI) curve (at −20 mV) of multiple mouse TTX-R channels in the presence of 0.75 μM HpTx1 (unpaired two-tailed $t$ test, WT mice: $t_{12} = 4.09$, $P = 0.0015$, $n = 7$; Na$_v$1.7-KO mice: $t_{20} = 6.447$, $P < 0.00001$, $n = 11$; Na$_v$1.8-KO mice: $t_{22} = 5.905$, $P < 0.00001$, $n = 12$; Nav1.9-KO mice: $t_6 = 1.999$, $P = 0.0925$, $n = 4$). Note that 1 μM TTX was applied in these experiments (**d–f**). **g** The dose–response curves for the HpTx1-induced inhibition of the fast inactivation of hNa$_v$1.9 expressed in ND7/23 cells ($n = 5$). The inset shows representative current traces (left) and normalized current traces (right) in the absence (black) and presence of 0.75 μM HpTx1 (red). **h** HpTx1 significantly slows the fast inactivation time of hNa$_v$1.9 ($n = 6$ for control, $n = 5$ for HpTx1, two-way ANOVA followed by Bonferroni's multiple comparisons test, treatment × voltage: $F_{(7,66)} = 6.386$, $P < 0.0001$; treatment: $F_{(7,66)} = 27.35$, ####$P < 0.0001$; voltage: $F_{(1,66)} = 411.9$, $P < 0.0001$). **i** Voltage dependence of the steady-state activation (G/G$_{max}$) and inactivation (I/I$_{max}$) of hNa$_v$1.9 for the control (black dots, $n = 5$ for activation, $n = 6$ for inactivation) and with 0.75 μM HpTx1 application (red diamonds, $n = 5$ for activation, $n = 9$ for inactivation). Data are presented as the mean ± S.E.M. *$P < 0.05$, **$P < 0.01$, ***$P < 0.001$, ****$P < 0.0001$, NS not significant. Exact $P$ (**e, h-i**) are presented in Supplementary Data 1. Source data are provided as a Source Data file.Source Data file.

in the SSI curve in the presence of 0.75 μM HpTx1 (Fig. 3f). These effects were limited to DRG neurons expressing Na$_v$1.9 (Fig. 3f; Supplementary Fig. 2b), which was further confirmed in ND7/23 cells heterologously expressing Na$_v$1.8 or Na$_v$1.9. HpTx1 significantly enhanced human Na$_v$1.9 (hNa$_v$1.9) currents in ND7/23 cells (Fig. 3g; Supplementary Fig. 2c) and potently inhibited the fast inactivation of this channel (Fig. 3g, h), consistent with the effect of HpTx1 on TTX-R Na$_v$ currents in mouse DRG neurons. The half-maximum effective concentration (EC$_{50}$) of HpTx1 was determined to be 0.47 ± 0.08 μM (Fig. 3g). In contrast, HpTx1 did not affect rat Na$_v$1.8 (rNa$_v$1.8) currents expressed in ND7/23 cells (Supplementary Fig. 2d).

Further investigation showed that 0.75 μM HpTx1 had no effect on the voltage dependence of the steady-state activation of hNa$_v$1.9 (control: V$_{1/2}$ = −38.2 ± 3.6 mV and k = 8.9 ± 1.5 mV; HpTx1: V$_{1/2}$ = −35.2 ± 3.0 mV and k = 10.0 ± 0.9 mV; $n = 6$, $P > 0.05$). However, the voltage dependence of the SSI was significantly shifted by approximately 13.4 mV in the presence of 0.75 μM HpTx1 (control: V$_{1/2}$ = −69.9 ± 1.3 mV and k = −9.0 ± 0.7 mV; HpTx1: V$_{1/2}$ = −56.5 ± 2.4 mV and k = −15.0 ± 0.6 mV; $n = 8$, $P = 0.0009$) (Fig. 3i; Supplementary Table 3). This result differed from the effect of HpTx1 on TTX-R Na$_v$ currents in DRG neurons, potentially due to differences between DRG neurons and ND7/23 cells and between mouse Na$_v$1.9 (mNa$_v$1.9)

and hNa$_v$1.9 sequences. Na$_v$1.9 channels in ND7/23 cells and DRG neurons have distinct posttranslational modifications and auxiliary subunits that are known to affect the pharmacological properties and physiological properties of Na$_v$ channels[35,36]. However, for both TTX-R Na$_v$ currents in DRG neurons and Na$_v$1.9 currents in ND7/23 cells, their window currents were obviously improved in the presence of HpTx1 (Fig. 3i). Indeed, 0.75 μM HpTx1 robustly increased the peak of the ramp current of TTX-R Na$_v$ in small DRG neurons from WT mice by 62.0 ± 8.8% (Supplementary Fig. 2e) and of hNa$_v$1.9 expressed in ND7/23 cells by 71.8 ± 14% (Supplementary Fig. 2f), potentially increasing Na$^+$ influx and leading to enhanced excitability of DRG neurons.

As a further test of toxin specificity, HpTx1 produced neither activation nor persistent inhibition when applied to other pain-related ion channels, including transient receptor potential vanilloid 1 (TRPV1) and acid-sensing ion channels (ASICs) (Supplementary Fig. 3). Taken together, these results suggested that HpTx1 enhances Na$_v$1.9 activity by inhibiting its fast inactivation and that the mechanism by which HpTx1 improves membrane excitability and evokes pain in Na$_v$1.7-KO mice might be related to this enhanced Na$_v$1.9 activity.

**HpTx1-triggered pain responses depend on Na$_v$1.9 activity**. We next tested the effects of HpTx1 on the membrane excitability of small DRG neurons from Na$_v$1.9-KO mice. As shown in Fig. 4a, 0.75 μM HpTx1 did not change RMP (control: −50.2 ± 1.7 mV; HpTx1: −49.1 ± 1.7 mV; $n = 29$, $P = 0.1828$), but significantly increased rheobase by 10.4 pA (control: 40.0 ± 4.3 pA; HpTx1: 50.4 ± 5.6 pA; $n = 25$, $P = 0.008$). Furthermore, HpTx1 had no effect on AP amplitude (control: 110.6 ± 1.3 mV; HpTx1: 109.7 ± 1.7 mV; $n = 25$, $P = 0.3132$) or input resistance (Supplementary Table 1). The increased rheobase in the presence of HpTx1 led to an obvious suppression in the evoked AP firing frequency of small DRG neurons from Na$_v$1.9-KO mice with HpTx1 treatment (Fig. 4b, c, $n = 25$). These effects differed from those observed in WT and Na$_v$1.7-KO mice, and these differences could be interpreted to be due to the inhibition of Na$_v$1.7 currents by HpTx1.

In further in vivo experiments, the pain-related behaviors induced by injection of HpTx1 into hind paws observed in WT mice were not observed in Na$_v$1.9-KO mice (Fig. 4d). In contrast to the decrease in thermal and mechanical stimulus thresholds induced by HpTx1 in WT mice, HpTx1 had an analgesic effect in Na$_v$1.9-KO mice. In Na$_v$1.9-KO mice, HpTx1 significantly increased the mechanical threshold (Fig. 4e) and the latency of paw withdrawal under noxious heat stimulus (Fig. 4f), most likely by inhibiting Na$_v$1.7.

Our data demonstrated that HpTx1 enhanced the excitability of primary afferent neurons and elicited pain, depending on the expression of Na$_v$1.9 and that Na$_v$1.9 activation might be required for such an effect. In addition, Na$_v$1.9 expression in DRG neurons from WT mice showed an ~59.6% overlap with the expression of IB4, a marker for small, unmyelinated nonpeptidergic fibers (Supplementary Fig. 4). As mentioned above, 50–65% of small DRG neurons from WT, fNa$_v$1.7 or Na$_v$1.7-KO mice exhibited decreased current thresholds for AP firing. These neurons may be the small nonpeptidergic neurons that express Nav1.9 and may transmit nociceptive signals from the periphery to the spinal dorsal horn[37].

**HpTx1 fails to affect pain responses in Na$_v$1.8-KO mice**. Given the important roles of Na$_v$1.8 in AP generation and pain signaling, we sought to determine whether Na$_v$1.8 is also required for HpTx1-induced pain, although HpTx1 did not directly affect Na$_v$1.8. We investigated the effects of HpTx1 on Na$_v$1.8-KO and Na$_v$1.7/Na$_v$1.8 double-knockout (Na$_v$1.7/Na$_v$1.8-DKO) mice (simultaneous knockout of Na$_v$1.7 and Na$_v$1.8). As shown in Fig. 5a–d, 0.75 μM HpTx1 depolarized the RMP of small DRG neurons from Na$_v$1.8-KO mice by 2.6 mV (control: −46.9 ± 1.9 mV; HpTx1: −44.3 ± 1.8 mV; $n = 18$, $P = 0.0037$) without affecting rheobase (control: 40.0 ± 6.1 mV; HpTx1: 38.9 ± 8.0 mV; $n = 18$, $P > 0.999$), AP amplitude (control: 83.4 ± 3.2 mV; HpTx1: 82.5 ± 2.3 mV; $n = 18$, $P = 0.9816$) or firing frequency (Supplementary Table 1). In Na$_v$1.7/Na$_v$1.8-DKO mice, 0.75 μM HpTx1 significantly depolarized the RMP of small DRG neurons by 5.2 mV (control: −52.3 ± 1.4 mV; HpTx1: −47.1 ± 1.2 mV; $n = 19$, $P < 0.0001$) and decreased rheobase by 11.6 pA (control: 60.0 ± 6.3 pA; HpTx1: 48.4 ± 6.4 pA; $n = 19$, $P = 0.0112$) but had no effect on AP amplitude (control: 88.4 ± 2.3 mV; HpTx1: 87.0 ± 2.5 mV; $n = 19$, $P = 0.5522$) or firing frequency (Fig. 5a–c, e; Supplementary Table 1). These data suggested that HpTx1 failed to enhance small DRG neuron AP firing in Na$_v$1.8-KO and Na$_v$1.7/Na$_v$1.8-DKO mice, despite depolarizing the DRG neuron RMP in both mouse lines.

Intraplantar injection of 10 μM HpTx1, a concentration that elicited acute pain in WT and Na$_v$1.7-KO mice, failed to affect the pain responses in Na$_v$1.8-KO mice (Fig. 5f). In addition, in contrast to the effects of HpTx1 in WT and Na$_v$1.7-KO mice, HpTx1-induced mechanical allodynia (Fig. 5g) and thermal hyperalgesia were abrogated in Na$_v$1.8-KO mice (Fig. 5h). Together, these data suggested that Na$_v$1.8 was also required for HpTx1-induced pain in mice.

**The mechanism of HpTx1 acting on Na$_v$1.7 and Na$_v$1.9**. At least six different neurotoxin receptor sites have been identified on Na$_v$ channels so far, with site 3 and site 4 being hot spots for spider peptide toxins[28,38]. The involvement of the domain IV (DIV) s3b-s4 segment in the formation of receptor site 3 is crucial for fast inactivation[22,39,40]. In this study, our data showed that HpTx1 slowed the development of the fast inactivation of Na$_v$1.9 (Fig. 3g, h), implying that HpTx1 might target site 3. To identify the region of Na$_v$1.9 critical for the toxin-induced inhibition of fast inactivation, we constructed several chimaeric channels. Because Na$_v$1.8 was resistant to HpTx1, we made a chimera, in which the voltage-sensor domain DIV (DIV s1–s4) of Na$_v$1.9 was replaced with the corresponding domain of Na$_v$1.8 (named Na$_v$1.9/1.8 DIV s1–s4) (Supplementary Fig. 5a). In this chimaera channel, HpTx1 exhibited a significant decrease in efficacy (Supplementary Fig. 5b). Further investigation showed that replacing the DIV s3b-s4 P1 region of Na$_v$1.9 abolished the effects of HpTx1 on the channel (Fig. 6a, b; Supplementary Fig. 5c), but replacing the DIV s1-s2 linker did not (Supplementary Fig. 5d). Moreover, we confirmed that reverse construction of Na$_v$1.9 DIV s3b-s4 P1 into Na$_v$1.8 (Na$_v$1.8/1.9 DIV s3b-s4 P1) conferred the efficacy of HpTx1 (Fig. 6b; Supplementary Fig. 5e). These results indicated that HpTx1 might inhibit the fast inactivation of Na$_v$1.9 through binding to the DIV s3b-s4 region. In addition, site-directed mutagenesis analysis, in which a total of seven amino acid residues in DIV s3–s4 P1 of Na$_v$1.9 were replaced with the corresponding residues of Na$_v$1.8, showed that three mutations (T1444L, E1450L, and N1451K) significantly reduced the toxin activity (Fig. 6c; Supplementary Fig. 5f–h), suggesting that these residues might directly affect the interaction of Na$_v$1.9 with HpTx1.

HpTx1 inhibited Na$_v$1.7 current amplitude without altering steady-state activation and inactivation (Fig. 3b). However, as shown in Supplementary Fig. 5i, progressively longer strong depolarization (to +80 mV) led to an increase in the fraction of Na$^+$ currents recovered from inhibition by HpTx1, and increased depolarization potentials were correlated with increased

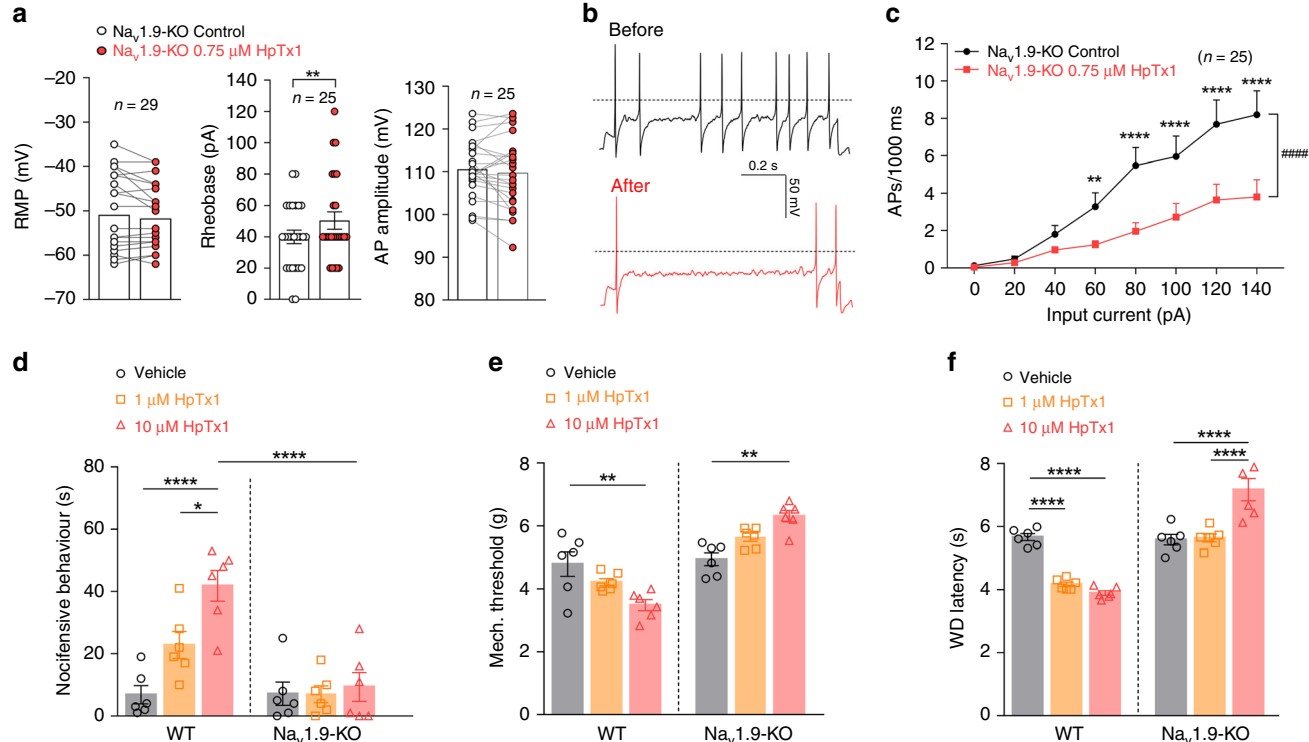

**Fig. 4 HpTx1-evoked pain hypersensitivity relies on Na$_v$1.9 activity. a–c** Current-clamp recordings show that HpTx1 decreases the membrane excitability of small DRG neurons from Na$_v$1.9-KO mice. **a** Bars show no significant changes in RMP (left, $n = 29$) or AP amplitude (right, $n = 25$), but a significant increase in rheobase (middle, $n = 25$, nonparametric Wilcoxon matched-pair signed-rank two-tailed test: $P = 0.008$) in the presence of 0.75 μM HpTx1. **b** AP traces recorded from a representative small Na$_v$1.9-KO DRG neuron before (black) and after (red) application of 0.75 μM HpTx1. The dashed lines indicate 0 mV. **c** Statistics plots show significant decreases in AP spike number in the presence of 0.75 μM HpTx1 ($n = 25$, two-way repeated measures ANOVA followed by Bonferroni's multiple comparisons test, treatment × inject current: $F_{(7,168)} = 8.834$, $P < 0.0001$; treatment: $F_{(1,24)} = 25.49$, ####$P < 0.0001$; inject current: $F_{(7,168)} = 25.28$, $P < 0.0001$). **d** Comparison of nocifensive behaviors (licking or biting) following intraplantar injection of vehicle (10 μl 0.9% saline, $n = 6$) versus HpTx1 (1 μM or 10 μM in 10 μl saline, $n = 6$) (two-way ANOVA followed by Tukey's multiple comparisons test, treatment × genotype: $F_{(2,30)} = 8.551$, $P = 0.0012$; treatment: $F_{(2,30)} = 11.04$, $P = 0.0003$; genotype: $F_{(1,30)} = 24.37$, $P < 0.0001$). **e** Mechanical response thresholds measured in paws in response to vehicle (black circles, $n = 6$), 1 μM HpTx1 (yellow squares, $n = 6$) or 10 μM HpTx1 (red triangles, $n = 6$) injections (two-way ANOVA followed by Tukey's multiple comparisons test, treatment × genotype: $F_{(2,30)} = 18.68$, $P < 0.0001$; treatment: $F_{(2,30)} = 0.0356$, $P = 0.9651$; genotype: $F_{(1,30)} = 67.3$, $P < 0.0001$). **f** Latency of WD to noxious heat stimuli measured after intraplantar injection of vehicle (black circles, $n = 6$), 1 μM HpTx1 (yellow squares, $n = 6$) or 10 μM HpTx1 (red triangles, $n = 6$) (two-way ANOVA followed by Tukey's multiple comparisons test, treatment × genotype: $F_{(2,30)} = 44.54$, $P < 0.0001$; treatment: $F_{(2,30)} = 9.701$, $P = 0.0006$; genotype: $F_{(1,30)} = 113.5$, $P < 0.0001$). All DRG neurons recorded were held at $-53 \pm 2$ mV. Data are presented as the mean ± S.E.M. *$P < 0.05$, **$P < 0.01$, ***$P < 0.001$, ****$P < 0.0001$. Exact $P$ (**c**–**f**) are presented in Supplementary Data 1. Source data are provided as a Source Data file.Source Data file.

dissociation. The data indicated that the binding of HpTx1 was reversed by prolonged strong depolarization that activated the voltage sensor, similar to our previous results with HWTX-IV[41] and HNTX-III[42], which are site 4 toxins that interact with Na$_v$1.7 domain II s3b-s4 (DII s3b-s4). Accordingly, chimaera channels of Na$_v$1.7 were constructed. A substitution of the DII s3b-s4 region of Na$_v$1.7 with the corresponding region of Na$_v$1.8 (Na$_v$1.7/1.8 DII s3b-s4) caused this channel to be completely insensitive to HpTx1 (Fig. 6d–f). Similarly, site-directed mutagenesis analysis indicated that D816 and E818 observably affected the efficacy of the HpTx1-Na$_v$1.7 interaction (Fig. 6e, f). In particular, mutating E818 to arginine, an oppositely charged amino acid residue located in the corresponding position of Na$_v$1.9, resulted in an ~20-fold reduction in HpTx1 activity (Fig. 6e, f), potentially explaining why HpTx1 did not depress Na$_v$1.9 currents through binding to DII s3-s4. In addition, two mutants of Na$_v$1.7 (F813S and G819S) increased the efficacy of HpTx1 (Fig. 6e, f). Furthermore, the reverse chimaera (Na$_v$1.8/Na$_v$1.7 DII s3b-s4) conferred toxin sensitivity with an IC$_{50}$ value of $3.0 \pm 0.6$ μM (Fig. 6e). However, site-directed mutants, in which seven amino acid residues in DII s3-s4 of Na$_v$1.8 were replaced with the

corresponding residues of Na$_v$1.7, were insensitive to HpTx1 (Supplementary Fig. 6). These results suggested that multiple amino acid residues in Na$_v$1.7 DII s3-s4 might be synergistically involved in the interaction with HpTx1.

## Discussion

In this study, we discovered that enhancing Na$_v$1.9 activity may recover the pain deficit in CIP caused by loss-of-function of Na$_v$1.7, which was revealed by the intensive study of HpTx1 as a probe. HpTx1 modulates Na$_v$ activities through an action mode distinct from previously reported peptide toxins, inhibiting Na$_v$1.7 and activating Na$_v$1.9, but not affecting Na$_v$1.8. Consequently, the effects of HpTx1 on pain responses depend on the expression of these three channels in small DRG neurons. HpTx1 causes pain in WT and Na$_v$1.7-KO mice and analgesia in Na$_v$1.9-KO mice, but is ineffective in Na$_v$1.8-KO mice. The results at the channel level and the data at the animal phenotype level are well connected by the action of HpTx1 on the membrane excitability of small DRG neurons. In other words, this interesting discovery establishes a link among Na$_v$1.7, Na$_v$1.8 and Na$_v$1.9 in the pain

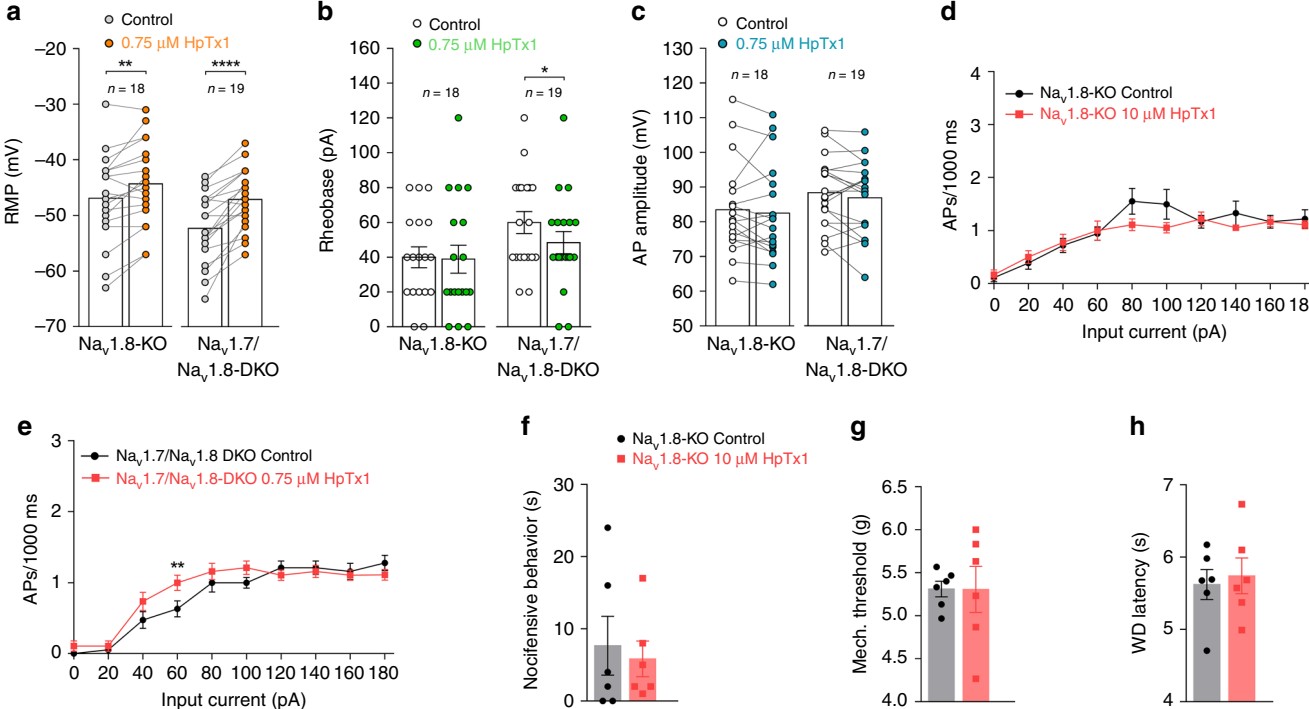

**Fig. 5 HpTx1 has no effect on pain responses in Na$_v$1.8-KO mice. a–e** Current-clamp recordings show the effects of HpTx1 on the excitability of small DRG neurons from Na$_v$1.8-KO mice and Na$_v$1.7/Na$_v$1.8-DKO mice. **a** RMP (two-way repeated measures ANOVA followed by Bonferroni's multiple comparisons test, treatment × genotype: $F_{(1,35)} = 5.792$, $P = 0.0215$; treatment: $F_{(1,35)} = 52.44$, $P < 0.0001$; genotype: $F_{(1,35)} = 3.389$, $P = 0.0741$), **b** rheobase (two-way repeated measures ANOVA followed by Bonferroni's multiple comparisons test, treatment × genotype: $F_{(1,35)} = 3.465$, $P = 0.0711$; treatment: $F_{(1,35)} = 5.092$, $P = 0.0304$; genotype: $F_{(1,35)} = 2.656$, $P = 0.1121$), and **c** AP amplitude (Na$_v$1.8-KO mice, $n = 18$; Na$_v$1.7/Na$_v$1.8-DKO mice, $n = 19$). **d, e** HpTx1 has no effect on the AP firing frequency of small DRG neurons from Na$_v$1.8-KO mice (**d**, $n = 18$) or Na$_v$1.7/Na$_v$1.8-DKO mice (**e**, $n = 19$, two-way repeated measures ANOVA followed by Bonferroni's multiple comparisons test, treatment × inject current: $F_{(7,126)} = 3.141$, $P = 0.0043$; treatment: $F_{(1,18)} = 4.275$, $P = 0.0534$; inject current: $F_{(7,126)} = 79.17$, $P < 0.0001$). **f–h** 10 μM HpTx1 (red diamonds) has no effect on nocifensive behaviors (**f**, $n = 6$) or mechanical (**g**, $n = 6$) or thermal pain (**h**, $n = 6$). All DRG neurons recorded were held at −53 ± 2 mV. Data are presented as the mean ± S.E.M. *$P < 0.05$, **$P < 0.01$, ****$P < 0.0001$. Exact $P$ (**a, b, e**) are presented in Supplementary Data 1. Source data are provided as a Source Data file.Source Data file.

signaling pathway and provides a helpful strategy for therapeutic development for Na$_v$1.7-related CIP.

Our study revealed that the activity of Na$_v$1.9 determines the responses of pain or no pain in mice treated with HpTx1. Enhanced activation of Na$_v$1.9 led to hyperexcitability of some small DRG neurons and pain responses in mice regardless of whether Na$_v$1.7 was lacking or inhibited by HpTx1, but Na$_v$1.8 was required in this process. HpTx1 evoked robust pain responses and produced profound hypersensitivity to mechanical and thermal stimuli in Na$_v$1.7-KO mice (Fig. 1c–e), as revealed by the depolarized RMP, reduced rheobase and increased AP firing in some small DRG neurons with Na$_v$1.9 activation by HpTx1 (Fig. 2e–g). In contrast, HpTx1 failed to induce pain in Na$_v$1.9-KO mice, but rather decreased DRG neuron excitability, blocked neuronal signaling, and induced analgesia (Fig. 4), consistent with Na$_v$1.7 inhibition by HpTx1. These observations demonstrate that the two phenotypes derived from the dual activities of HpTx1 (inhibiting Na$_v$1.7 and activating Na$_v$1.9) are manifested in these two mouse mutants. However, WT mice, which express Na$_v$1.7 and Na$_v$1.9, displayed pain responses with HpTx1 treatment, indicating that enhanced Na$_v$1.9 activity might conceal the analgesic activity derived from the inhibition of Na$_v$1.7. Our data suggest that the inhibition of Na$_v$1.7 by HpTx1 treatment may be considered analogous to KO of Na$_v$1.7 to some extent. One possible explanation for the ability of HpTx1-induced Na$_v$1.9 activation to overshadow the analgesic activity derived from HpTx1-induced Na$_v$1.7 inhibition based on the role of Na$_v$1.7 and Na$_v$1.9 as AP threshold channels[18,20,43,44] is that HpTx1-bound

Na$_v$1.9 largely depolarized the RMP of DRG neurons, potentially compensating for the absence of Na$_v$1.7 and leading to AP generation. Studies have proven that a moderately depolarized RMP contributes to the decrease in the current threshold for AP generation by promoting the opening of other ion channels (i.e., Na$_v$1.8 and voltage-gated calcium channels)[45,46]. Our data demonstrated that HpTx1 depolarized RMP by 2.0 mV and 3.2 mV in WT and Na$_v$1.7-KO mice, respectively, leading to a reduced current threshold and increased AP and finally enhancing the membrane excitability of some small DRG neurons (Supplementary Table 1). Some studies have shown that histamine induces short-range AP propagation in sensory terminals and evokes a local neurogenic flare in individuals with CIP[6,47], which suggests that nociceptive afferents lacking Na$_v$1.7 are not entirely nonfunctional. Histamine has been found to improve the activity of Na$_v$1.9 and enhance AP firing in DRG neurons[37,48,49]. Studies examining the mechanism by which a gain-of-function mutation of Na$_v$1.9 causes painful disorders have shown depolarized RMP in DRG neurons expressing some of these mutants (for example, a significant 3.5-mV depolarization of RMP was reported for the L1158P mutant)[50–52]. On the other hand, Na$_v$1.9 does not contribute much to the amplitude of APs[44], suggesting that enhancing Na$_v$1.9 activity is unable to compensate for the loss of Na$_v$1.8, which contributes to the majority (58–90%) of the inward current during the rising phase of an all-or-none AP in nociceptive sensory neurons[19,47,53]. Indeed, our data showing that HpTx1 failed to affect AP firing in Na$_v$1.8-KO DRG neurons and pain sensitivity in Na$_v$1.8-KO mice are consistent with this

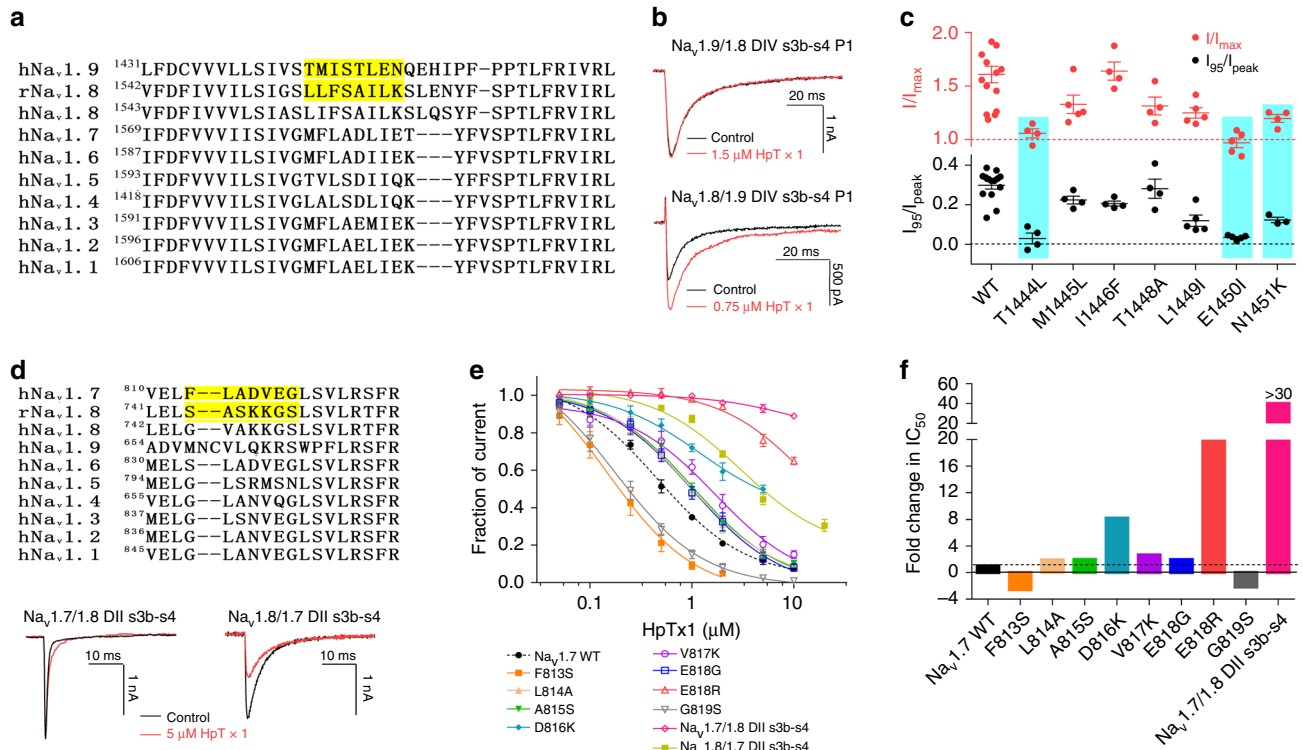

**Fig. 6 The molecular mechanism of HpTx1 action on Na_v1.9 and Na_v1.7 channels. a** Sequence alignments corresponding to the DIV s3b-s4 region of Na_v subtypes. The highlighted sequences show the regions swapped between Na_v1.8 and Na_v1.9. **b** Representative current traces from Na_v1.9/1.8 DIV s3b-s4 P1 (top) and Na_v1.8/1.9 DIV s3b-s4 P1 (bottom) chimaera channels in the absence (black) and presence (red) of HpTx1. **c** Effects of HpTx1 on WT and mutant hNa_v1.9 channels. Dot plots display the effect of 0.75 μM HpTx1 on the peak current (top, $n = 14$ for WT; $n = 4$ for T1444L, M1445L, I1446F, and T1448A; $n = 5$ for L1449I and E1450L; $n = 3$ for N1451K) and the persistent current (bottom, $n = 14$ for WT; $n = 4$ for T1444L, I1446F, T1448A, and N1451K; $n = 5$ for M1445L, L1449I, and E1450L). Key residues involved in the interaction between HpTx1 and hNa_v1.9 are labeled (one-way ANOVA with Dunnett's multiple comparison test, $I_{95}/I_{peak}$: $F_{(7,35)} = 17.72$, $P < 0.0001$; $I/I_{max}$: $F_{(7,38)} = 8.157$, $P < 0.0001$). **d** (top) Sequence alignments corresponding to the DII s3b-s4 region of Na_v subtypes. The highlighted sequences show the regions swapped between Na_v1.7 and Na_v1.8. Representative current traces from Na_v1.7/1.8 DII s3b-s4 (bottom left) and Na_v1.8/1.7 DII s3b-s4 (bottom right) chimaera channels in the absence (black) or presence of 5 μM HpTx1 (red). **e** Dose-dependent inhibitory curves show the effect of HpTx1 on WT ($n = 7$) and mutant hNa_v1.7 channels ($n = 4$ for F813S, $n = 6$ for L814A and A815S, $n = 3$ for D816K, $n = 6$ for V817K, $n = 7$ for E818G, $n = 4$ for E818R, $n = 5$ for G819S and $n = 3$ for Na_v1.7/1.8 DII s3b-s4) and the Na_v1.8/1.7 DII s3b-s4 chimaera channel ($n = 5$). **f** Bars show the fold changes in $IC_{50}$ values of HpTx1 for mutant channels compared with that for the WT hNa_v1.7 channel. Data are presented as the mean ± S.E.M. Exact $P$ (**c**) are presented in Supplementary Data 1. Source data are provided as a Source Data file.Source data file.

explanation (Fig. 5). In addition, Rush et al.[45] demonstrated that the functional consequence of depolarized RMP in part depends on the Na_v1.8 expression profile of affected neurons. Therefore, another possible explanation is that depolarization of the RMP of DRG neurons lacking Na_v1.8 might result in inactivation of some other channels, resulting in significantly fewer active channels to induce AP electrogenesis.

Our study also provides pharmacological insight into the relationship of Na_v1.7, Na_v1.8, and Na_v1.9 in the pain signaling pathway. According to our findings and previous data, the three channels may have different roles in the regulation of membrane excitability and AP firing in DRG neurons. AP generation is the process of neuron membrane depolarization, during which the membrane potential can be described to reach three levels: the RMP, threshold, and upstroke[47,54]. Na_v1.7, Na_v1.8, and Na_v1.9 play distinct roles in these three levels, and their collaboration is required for the generation of a complete AP and repeated firing[47]. Na_v1.9 is believed to set the RMP level of DRG neurons[44], supported by the ability of HpTx1-induced Na_v1.9 activation and pain-inducing gain-of-function Na_v1.9 mutations to depolarize the RMP of DRG neurons[50–52]. Our data demonstrated that the enhanced activity of Na_v1.9 reduces the current threshold, showing that, similar to Na_v1.7, Na_v1.9 can also function as a gain amplifier to amplify subthreshold stimuli in AP firing[18,20,43,44].

Multiple studies have indicated that Na_v1.8 may serve as a major contributor to the rising phase of the AP[19,53]. Recently, Bennett et al. further refined the contribution of Na_v1.7 to AP generation[47], proposing that Na_v1.7 also contributes to the rising phase, consistent with our finding of a slight decrease in AP amplitude when Na_v1.7 is knocked out (Supplementary Table 1). To date, no evidence has shown that Na_v1.9 contributes to the rising phase. These findings suggest that the roles of the three channels in AP firing overlap to some extent. Under normal conditions, because of the low amount of Na_v1.9 expressed in DRG neurons, its contribution to AP firing related to pain signaling may not be prominent[55]. Unlike for Na_v1.7 and Na_v1.8, no loss-of-function mutants for Na_v1.9 have been found in the clinic, and Na_v1.9-KO mice do not show evident changes in physiological nociception. On the other hand, in hyperalgesic conditions, for example, with HpTx1 treatment, inflammatory factors[56], cold[51,57] and gain-of-function mutations[16,50–52,58], the contribution of Na_v1.9 can be enhanced and even compensate for the loss of Na_v1.7. Therefore, enhanced activation of Na_v1.9 may recover the pain responses in Na_v1.7-related CIP, as revealed in our studies.

Potentiated pain sensitivity should be beneficial for human Na_v1.7-null CIP individuals. Minett et al.[59] observed that upregulated endogenous opioids contribute to the analgesia phenotype in Na_v1.7-null mutant mice and that the analgesia associated

with the loss of Na$_v$1.7 in both mice and humans is substantially reversed by the opioid antagonist naloxone. Our study also revealed that pain sensitivity in Na$_v$1.7-related CIP can be evoked by activating Na$_v$1.9. These studies confirm that a practical treatment strategy for Na$_v$1.7-related CIP may be through targeting the different pain signaling pathways other than Na$_v$1.7 itself. In addition to the ability of agonists such as the peptide toxin HpTx1 to enhance Na$_v$1.9 activity, various other factors have been shown to affect Na$_v$1.9 activity as well. For example, glial cell-derived neurotrophic factor (GDNF) significantly increases Na$_v$1.9 mRNA and current density[60], the endogenous molecule contactin promotes the surface expression of Na$_v$1.9[61,62], and some inflammatory mediators, such as prostaglandin E2 (PGE2), GTP, and histamine, can markedly increase the current density of Na$_v$1.9 currents and AP firing[37,48,49,63,64]. Therefore, enhancing the activation of Na$_v$1.9 by regulating these endogenous molecules may be a feasible strategy for treating CIP. However, the possible risks caused by activating Na$_v$1.9 should be noted. Similar to some gain-of-function mutations of Na$_v$1.9, activated Na$_v$1.9 might result in spontaneous pain in humans[16,50–52], while hyperactive Na$_v$1.9, similar to the L811P and L1302F mutants, may lead to pain insensitivity[3,46].

## Methods

**Venom collection and toxin purification**. The venom was obtained by electrical stimulation of female spiders of *H. venatoria*, and the freeze-dried crude venom was stored at −20 °C prior to analysis. Lyophilized venom was dissolved in double-distilled water. Every time, 10 mg dried venom was purified by semipreparative reverse-phase HPLC using an Ultimate® XB-C18 column (300 Å, 10 mm × 250 mm, Welch Materials Inc., Shanghai, China) on the Hanbon HPLC system (Hanbon Sci&Tech., Jiangsu, China). The following linear gradient of solvent A (0.1% formic acid in acetonitrile) in solvent B (0.1% formic acid in water) was used at a flow rate of 3 ml min$^{-1}$: 15% A for 5 min, then 15–60% A over 45 min. Absorbance was measured at 215 nm, and fractions were collected and lyophilized before storage at −20 °C. The target fraction was subjected to the second round of RP-HPLC (Waters alliance 2695 HPLC system) using a XB-C18 column (300 Å, 4.6 mm × 250 mm, Welch Materials Inc., Shanghai, China) with a slower increasing acetonitrile gradient (acetonitrile at an increasing rate of 0.5% per minute, and a flow rate of 1 ml min$^{-1}$) to obtain the purified HpTx1.

**Mass spectrometry and sequencing**. The molecular weight of a peptide was analyzed by MALDI–TOF-TOF MS spectrometry (AB SCIEX TOF/TOF$^{TM}$ 5800 system, Applied Biosystems, USA). The entire amino acid sequence of a peptide was obtained by automated Edman degradation using an Applied Biosystems 491 pulsed-liquid-phase sequencer from Applied Biosystem Inc.

**Plasmid constructs and mutagenesis**. Human Na$_v$1.7 (hNa$_v$1.7) and rat Na$_v$1.8 (rNa$_v$1.8) clones and beta subunit (β1 and β2) clones were kindly gift from Dr. Theodore R.Cummins (Department of pharmacology and Toxicology, Stark Neurosciences Research Institute, Indiana University School of Medicine, USA). hNa$_v$1.7 and rNa$_v$1.8 were subcloned into the vectors pcDNA3.1 and pCMV-blank vectors, respectively. hNa$_v$1.9 was subcloned into the pEGFP-N1 vector. The C-terminal of hNa$_v$1.9 was linked a GFP to construct a fusion protein channel (hNa$_v$1.9-GFP)[48]. The detail methods of mutagenesis are provided in the Supplementary Methods.

**Cell culture and transfection**. ND7/23 and HEK293T cells were maintained in Dulbecco's modified Eagle's medium (DMEM) supplemented with 10% fetal bovine serum, 2 mM L-glutamine, 100 U ml$^{-1}$ penicillin and 100 μg ml$^{-1}$ streptomycin in a 5% CO$_2$ incubator at 37 °C. Cells were trypsinized, diluted with culture medium, and grown in 35-mm dishes. When grown to 90% confluence, ND7/23 cells were transfected with hNa$_v$1.9-GFP or hNa$_v$1.9-GFP mutants using the transfection kit X-tremeGENE HP DNA Transfection Reagent (Roche, Basel, Switzerland) according to the manufacturer's instructions. Transfected cells were first maintained at 37 °C with 5% CO$_2$ for 24 h, and then incubated at 29 °C with 5% CO$_2$ for 20 h before use in electrophysiology experiments. Transfections of hNa$_v$1.7 and hNa$_v$1.7 mutants together with β1 and β2-eGFP and other ion channels (K$_v$4.2, TRPV1 and ASICs with eGFP) into HEK293T cells, and rNa$_v$1.8 and rNa$_v$1.8 mutants together with eGFP into ND7/23 cells were performed by using Lipofectamine 2000 (Thermo Fisher Scientific), according to the manufacturer's instruction. Six hours after transfection, the cells were seeded onto poly-D-lysine-coated coverslips (Thermo Fisher Scientific) and maintained at 37 °C in 95% CO$_2$ for 24 h before whole-cell patch-clamp recording. The green fluorescent was used for visual identification of individual transfected cells.

**Dorsal root ganglion neuron isolation and culture**. Six to eight-week-old C57BL6 WT, fNa$_v$1.7, Na$_v$1.7-KO, Na$_v$1.8-KO Na$_v$1.7/Na$_v$1.8-DKO, or Na$_v$1.9-KO mice were euthanized via cervical dislocation under anesthesia. Dorsal root ganglion (DRG) neurons were collected from the lumbar spinal cord L4–L5. DRG neurons were dissociated by enzymatic treatment with collagenase (1 mg ml$^{-1}$) and trypsin (0.3 mg ml$^{-1}$) at 37 °C for 30 min. cells were seeded onto poly-L-lysine-coated coverslips and cultured in DMEM (Gibco) containing 10% heat-inactivated fetal bovine serum (Gibco) and at 37 °C in a humidified incubator with 5% CO$_2$ for 3 h before whole-cell patch-clamp recording. Note that male and female mice in half were used in each experiment.

**Electrophysiology**. Whole-cell patch-clamp recordings were performed at room temperature (25 ± 2 °C) using an EPC-10 USB patch-clamp amplifier operated by PatchMaster software (HEKA Elektronik, Lambrecht, Germany) or Axopatch 200B amplifier (Molecular Devices). Fire-polished electrodes (2.0–2.5 MΩ) were fabricated from 1.5-mm capillary glass using a P-97 puller (Sutter, Novato, CA). Capacity transients were canceled; voltage errors were minimized with 80% series resistance compensation. The liquid junction potential was corrected using Axopatch 200B series amplifier, the Henderson equation was used to calculate the junction potential based on the ionic strength of bath and pipette solution. During the whole-cell recording performed in the EPC-10 USB amplifier, the liquid junction potential was not corrected. Voltage-dependent currents were acquired with Patchmaster at 5 min after establishing a whole-cell configuration, sampled at 30 kHz, and filtered at 2.9 kHz. The configuration of electrophysiology recording buffer and stimulation pulse are provided in the Supplementary Information.

For electrophysiology experiments, the stock solution of HpTx1 was diluted with fresh bath solution to a concentration of tenfold of the interested concentration, 30 μl of the concentrated peptide was diluted into the recording chamber (containing 270 μl bath solution) far from the recording pipet (the recording cell), and was mixed by repeatedly pipetting to achieve the specified final concentration. TTX and capsaicin were dissolved in DMSO to make 1 mM stock solutions. The final concentration of DMSO did not exceed 0.1%, which was found to have no significant effect on Na$^+$ currents.

**Animal experiments**. Na$_v$1.9-KO mice have been described earlier[56]. The floxed Na$_v$1.7 mice and Na$_v$1.8-Cre mice are described by Nassar et al.[30]. These mice were housed at the constant temperature of 24 °C and 50–60% humidity under controlled conditions of 12 h light/dark cycles and provided with free access to laboratory-standard food and water.

Seven to eight-week-old male ($n = 3$) and female ($n = 3$) mice were used on behavioral testing. Before testing, mice were habituated on an elevated platform of the mesh floor and plastic testing chambers for 30 min. HpTx1 was dissolved in saline and administered at doses of 1 μM and 10 μM, respectively. HpTx1 was injected into the hind paws of mice (10 μl saline, 1 μM or 10 μM HpTx1), and the seconds of licking/biting behavior were immediately recorded during a 20 min period following the injection. Nocifensive responses were assessed by the total seconds of licking/biting behavior during 20 min.

The latency to respond to heat was measured to assess the thermal pain threshold after 30 min of injection (Planter Test Analgesia Meter, IITC Inc. Life Science). It was recorded three times by using radiant light heat onto the plantar side of the paw with an interval of at least 5 min. In order to avoid injury to the mice, a cutoff of 20 s was set.

The mechanical withdrawal threshold was assessed by recording the max force that was continuously applied until paw withdrawal (Electronic von Fery Anesthesiometer, IITC Inc. Life Science), and recorded three times with an interval of at least 5 min. Note that all of the behavioral tests were double blind.

**Immunofluorescence**. DRG neurons were collected from the lumbar spinal cord (L4–L5). The DRG were fixed in 4% PFA in PBS for 2 h 30 min at 4 °C then cryoprotected in PBS containing 30% sucrose overnight at 4 °C. The DRG were frozen in OCT, and cryosectioned at 12 μm. The sections were permeabilized in PBS containing 0.5% TritonX-100 for 10 min, and were blocked with 10% goat serum for 1 h. The sections were incubated for 24 h at 4 °C with polyclonal rabbit anti-Na$_v$1.9 (1:200; alomone labs). Then the sections were incubated with biotinylated griffonia simplicifolia Lectin I isolectin B4 (10 μg ml$^{-1}$, Vector Laboratories, California, USA) for 30 min at room temperature. After incubation, sections were washed three times for 5 min in PBST (0.05% Tween20). Finally, incubated with Alexa-Fluor 488-nm (1:500, Invitrogen) and Alexa-Fluor 594 Streptavidin (1:200, Yeasen Biotechnology, Shanghai, China). After being washed three times for 5 min in PBST, sections were mounted with coverslips. Fluorescence images were acquired with the FV1000 confocal microscope (Olympus, Tokyo, Japan).

**Study approval**. All of the animal experiments were performed in accordance with the Guidelines for Laboratory Animal Research set by Hunan Normal University and Huazhong University of Science and Technology. The experiments were approved by the Institutional Animal Care and Use Committee of the College of Medicine, Hunan Normal University and Ethics Committee of Huazhong University of Science and Technology.

**Data analysis**. Data were analyzed with PatchMaster v2x73 (HEKA Elektronik), Clampfit 9 (Version 9.2.0.09, AXON), IgorPro6 (Version 6.1.0.9, WaveMetrics, Lake Oswego, OR, USA), Prism 7 (Version 7.00, GraphPad Software), FV10-ASW (Version 01.07.03.00, Olympus, Japan) and Office Excel 2010 (Version 14.0.4760.1000, Microsoft, USA). All values are shown as mean ± S.E.M., and $n$ represents the number of animals or cells examined. One-way ANOVA and two-way ANOVA were used to assess the difference between multiple groups. Two grouped data were analyzed by the Kolmogorov–Smirnov normality test before $t$ test analysis. If the data were normally distributed, a parametric $t$ test was used. Otherwise, a nonparametric $t$ test was used. In figure legends, statistical method to a specific experiment is mentioned, and the F, t, df (degree of freedom) and $p$-values are also shown. Significant levels were set at $p < 0.05$ and the exact $p$-values are presented in Supplementary Data 1. Statistical analyses were performed with Prism 7 (Version 7.00, GraphPad) software.

**Reporting summary**. Further information on research design is available in the Nature Research Reporting Summary linked to this article.

## Data availability

The authors declare that all data supporting the findings of this study are available in the article and its Supplementary Information Files, or on request from the corresponding author. The source data underlying Fig. 1a, c–e, g, h, Fig. 2a–e, g, Fig. 3a, b, g–i, Fig. 4a, d–f, Fig. 5a–c, f–h, Fig. 6c and Supplementary Figs. 1b–f, Fig. 2a–d, Fig. 3, Fig. 4, Fig. 5i, Fig. 6 are provided as a Source Data file.

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

## Acknowledgements

We thank Dr. Patrick Delmas (Université de la Méditerranée, Marseille Cedex, France) for providing Na$_v$1.9-KO mice, and Dr. Waxman SG (Yale University School of Medicine, West Haven, Connecticut, USA) for providing the floxed Na$_v$1.7 mice and Na$_v$1.8-Cre mice. This work was supported by funding from the National Science Foundation of China (31800655) and the China Postdoctoral Science Foundation funded project (2018M632968) to X.Z., the National Science Foundation of China (31570782, 31770832) and the Hunan Provincial Natural Science Foundation of China (14JJ1018) to Z.H.L., the National Science Foundation of China (31671301, 31871262) and the National Key R&D Program of China (2016YFC1306000) to J.Y.L., and the National Science Foundation of China (31872718) to S.P.L.

## Author contributions

X.Z. and T.B.M. conducted most of the experiments and analyzed the data, including peptide toxin purification, patch-clamp recording, mutagenesis, immunofluorescence, animal behavior tests, and data analysis; L.Y.Y. also conducted patch-clamp recording and animal behavior tests. L.L.L. conducted animal behavior tests. S.J.P., Z.Q.W., Z.X., Q.F.Z., L.W., and Y.Z.H. also conducted peptide toxin purification and patch-clamp recordings; M.Z.C. conducted MALDI–TOF mass spectrometry and data analysis; X.W.Z. designed the animal behavior tests. X.Z. and T.B.M. prepared the paper; X.Z., T.B.M., S.P.L., J.Y.L., and Z.H.L. designed the study and reviewed the paper.

## Competing interests

The authors declare no competing interests.
