## [Peer Review File · Nature Communications]

Reviewers' comments:

Reviewer #1 (Remarks to the Author):

The authors identify a novel venom-derived peptide that works to induce mechanical allodynia and pain sensitivity by activating Nav1.9, while also interestingly inhibiting Nav1.7. The site-directed mutagenesis experiments implicating site 3 or site 4 in the differential response of Nav1.9 and Nav1.7, respectively, was very convincing. However, the significance of this work is broader than this and two-fold. First, it presents a proof of principle for a new therapeutic avenue in the treatment of congenital insensitivity to pain by rescuing painlessness via a Nav1.9-mediated mechanism. Secondly, the authors have further refined and improved knowledge of the respective contributions of Nav1.7-9 to action potential electrogenesis.

Major comments

1) The discussion and implications of these results are interesting, but could be further strengthened by discussions of the following findings and papers.

a) Rush, A.M. et al, A single sodium channel mutation produces hyper- or hypoexcitability in different types of neurons. *Proc. Nat. Acad. Sci.*, 103: 8245-8250, 2006.

i) This paper demonstrates the principle that the effect of Nav1.7 mutations on excitability can depend on cell background

b) Rush, A.M., Waxman, S.G. PGE2 increases the tetrodotoxin-resistant Nav1.9 sodium current in mouse DRG neurons via G-proteins, *Brain Research*, 1023: 264-271, 2004.

i) Agents such as prostaglandins can influence Nav1.9 current

c) Rush, A.M. et al, Contactin regulates the current density and axonal expression of TTX-R but not TTX-S sodium channels in DRG Neurons, *Eur. J. Neurosci.*, 22:39-49, 2005.

i) Endogenous molecules such as contactin can influence level of expression of Nav1.9

2) In Nav1.8-KO mice, intraplantar injections of HpTx1 failed to affect nocifensive behavior.

Additionally, there was no change in mechanical threshold or withdrawal latency. This is interesting, because 1.8 KO mice theoretically express 1.7 and 1.9 like WT mice, but WT mice showed enhanced pain behavior in response to intraplantar injection of the toxin. Based on the ND7 cell data, the toxin does not directly interact with Nav1.8. The authors conclude that Nav1.9 could "largely compensate for the absence of Nav1.7 but not Nav1.8." Could the authors expand on this? A related but distinct explanation is that the RMP of 1.8-null DRG neurons is the most depolarized, inactivating Nav1.7 and the lack of Nav1.8 now results in significantly fewer active channels to induce action potential electrogenesis. This hypothesis is based on the findings of the paper cited in point 1a, above.

3) When neurons are lacking Nav1.9, the effect of the toxin is clearly to induce analgesia (increased current threshold, decreased AP firing, increased mechanical threshold and latency of paw withdrawal). When neurons are lacking Nav1.7, the effect of the toxin is clearly to induce pain-sensitivity (reduced mechanical threshold and reduced latency to paw withdrawal). WT mice, however, presumably expressing both Nav1.7 and -1.9 exhibited reduced threshold and latency. This seems to indicate that the effect on Nav1.9 predominates over the effect on Nav1.7. Could you hypothesize as to why this is?

Minor comments

1) Only 50% of WT small-diameter DRG neurons were HpTx1-sensitive (reduced current threshold, increased AP firing, etc.). Not all DRG neurons express Nav1.9 at high or appreciable levels. It would be interesting to note whether IB4-positivity was associated with the 50% of DRG neurons that responded to the toxin versus a lack of IB4-positive staining in nonresponsive cells. If no staining was conducted, could you tell, based off sample traces, if Nav1.9 currents were present in these non-responder neurons (assuming voltage-clamp was also done on these cells).

2) Building on that point, was a subgroup analysis conducted of the HpTx1 non-responding

neurons, comparing them to WT neurons injected with control? If the non-responders had minimal Nav1.9 currents, one would expect that these neurons might show an increased rheobase compared to WT neurons that were injected with control, especially since Nav1.9-KO neurons had an increase rheobase when exposed to the toxin.

3) Nav1.6 is also found within DRG neurons. In WT mouse DRG neurons, you found that 62.9% of TTX-S currents were inhibited by 1 micromolar HpTx-1. Was the effect of the toxin on Nav1.6 investigated? The data showing the inhibitory effect on Nav1.7 is strong on its own, but a small discussion on the Vasylyev et al 2014, which showed that roughly 70% of TTX-S current in mice, would lend further evidence to this reduction in TTX-S current being mediated by Nav1.7 and not Nav1.6.

4) In figure 2a, the HpTx1-induced calcium release looks significantly closer to background (almost unchanged, even) and significantly less like the positive-control. Figure 2b shows the average ratiometric calcium response increasing from 1 to approximately 1.05 after application. Is this significant?

Reviewer #2 (Remarks to the Author):

In this study entitled "Activating Nav1.9 recovers pain responses in Nav1.7-related CIP as revealed by a spider toxin", Zhou et al characterize the effects of HpTx1 on different types of sodium channels and, by using various Nav channel KO mice, characterize the effects of HpTX1 on pain behavior and DRG neuron excitability. There is a lot of work presented, and some of the results are interesting, but I have many reservations.

Major concerns.

1-The premise that congenital insensitivity to pain (CIP) could be treated using a toxin such as HpTX1 seems far-fetched. Leaving aside that not all forms of CIP are due to loss-of-function mutations of Nav1.7, this toxin causes spontaneous pain in addition to restoring sensitivity to painful stimuli. The latter effect is good (assuming it could be achieved in a sustained way without causing adverse effects... including via blockade of Kv4.2) but the former is obviously very bad. The results of the study are interesting without trying to find some clinical application that isn't well thought out.

2-The presentation of data and statistical analysis raise some significant concerns. In many cases the sample size is not reported, and when it is mentioned in a figure legend, it is often unclear which panel it refers to, and there are inconsistencies with the main text. There is no mention of whether the data are normally distributed and, therefore, whether parametric tests are appropriate. Individual data points are not shown on the majority of graphs, and so the reader cannot even start to judge this. Many of the answers provided on the Reporting Summary are demonstrably wrong.

3-In Figure 2b, the graph shows a single curve which is apparently the average response for 20 neurons. The curves for individual cells are not shown, nor are there any standard error markings around the group curve. Furthermore, I'm wondering if the authors have included all cells, or just cells that responded to the HpTX (this comes up again in my next point). 20 cells is very little for a calcium imaging experiment. Lastly, the figure legend refers to Fura-2, but the Methods discuss Fluo-2. Which is it?

4-On p 7, the authors explain that 15 out of 30 DRG neurons exhibited a reduced current threshold after HpTX. In Fig. 2c-f, does that mean that only data from HpTX-"responsive" cells are being reported. The figure legend does not provide the necessary information to resolve this confusion. If the authors have done what I think they've done, it's totally unacceptable to remove data from cells simply because they don't respond, and then claim that the responsive cells show a

significant response. A change in the distribution of the data (because of mixture of responsive and unresponsive cells) could be shown statistically (e.g. Kolmogorov-Smirnov test) and it would be worth trying to identify an independent criterion by which identify responsive vs unresponsive cells (e.g. peptidergic vs. non-peptidergic small DRG cells), but what the authors seem to have done is unacceptable. The same problem seems to apply to the latter half of Figure 2.

5-I think it is misleading when the authors write that the "identified" a spider peptide toxin called HpTX1. It is not until halfway through the Results (p 8) that the authors mention that this toxin was previously identified and is known to block certain potassium channels.

6-The author refer to extended figure 1e and f as evidence that HpTx1 does not have any effect on DRG potassium or calcium channels. But those panels only show a single sample trace showing no effect of the toxin. That is completely insufficient to show a lack of effect.

7-Were the behavioral tests blinded? This is mentioned in the reporting summary, but not in the manuscript. Were any of the electrophysiological tests blinded?

8-I am struck by the number of times review articles are cited when referring to specific findings. For instance, at the bottom of p 3, the authors explain the contribution of Nav1.8 and Nav1.7 to different parts of the action potential, each time citing a different review article. If this is a key issue – which I think it is – then the primary studies should be cited.

9-Was the liquid junction potential accounted for when reporting the data. On p 24, it was mentioned that liquid junction potential was zeroed before seal formation, but that does not remove the effect after breaking in and recording in whole-cell mode.

Other concerns:

-For Fig 3d, were all of those experiments conducted in TTX.

-*s are referred to later in the figure 3 legend, but none are shown on the figure.

-Extended data figure 3 refers to n=4, but only sample traces are shown on the figure.

Reviewer #3 (Remarks to the Author):

Main text is 6,459 words (Intro, Methods, Results, Discussion) – this needs to be considerably shortened (Per instructions to Authors, "main text of no more than 5,000 words"). The Introduction rambles and needs to be written and significantly shortened.

The English writing (grammar, noun-verb agreement, word choice, claims/statements of fact, etc.) is not acceptable in the current submission; it needs to be carefully reviewed and edited by a native speaker of English; examples include, but are by no means limited to:

Line 17 "nociceptor" should be "nociceptors"

Line 19 "loss" should be "lose"

Line 20-22 "So far, the therapeutic agents for treating the Nav1.7-related CIP are still in its infancy" – as written, this implies that such therapeutic agents exist (Minett et al., 2015); but this is possibly the only example, and caution should be used when citing it.

Line 24 "conformed" should be "confirmed"

Line 33 "and propose that Nav1.7, Nav1.8 and Nav1.9 are interrelated in pain signaling" – this "proposal" has long been put forth by Steve Waxman and his colleagues for over two decades – see recent review (Bennett, Clark, Huang, Waxman & Dib-Hajj, 2019).

And that is just in the Abstract; there are comparable problems throughout. It is the responsibility of the authors to have the manuscript in its entirety reviewed and revised appropriately.

Introduction:

Lines: 54-58 – clinical trials with Nav1.7 blockers – to merely state that trials have been undertaken is to minimize the point the authors are trying to make – the molecules actually show efficacy, but that point needs to be carefully conveyed (the study by (Cao et al., 2016), for example, only examined five subjects, and the effect was modest).

Line 63 “Urgent” – Nav1.7 loss-of-function mutations leading to congenital insensitivity to pain are extremely rare (Cox et al., 2006; Cox et al., 2010; McDermott et al., 2019); to say that a treatment is “urgently” needed is overstating the case. “needed” will suffice.

Lines 62-63, References 22,23,24 – discussion of animal venoms – all the cited references are to specific Nav1.1 blockers. Surely there are others and they should be cited.

Line 75 – “It produces a large window current” – this is not a “stand alone” sentence.

Methods/Results

Mice – “6-8 week-old adult C57Bl6” – at best these would be categorized as “young adult” but to avoid a semantic discussion, simply remove the word “adult”. The authors need to specify throughout the manuscript whether behavioral studies were sex (male:female)-balanced, and need to indicate sex of the mice used to provide DRG neurons. If only a single sex was used in either or both of these experiments, this will present significant problems with respect to understanding the significance of the observations given that profound sex-based differences are present when evaluating pain-related behaviors and their underlying mechanisms (Klein et al., 2015; Mogil, 2012).

Results

Line 104-105; “The venom fractions were screened for pain-induced activity.” – 15 crude venoms were fractionated – please indicate how many purified samples were obtained following fractionation. Of that number please confirm that all other purified fractions failed to elicit nocifensive behaviors.

Lines 123-127 – “Similar to control (fNav1.7) littermate mice, injection of 10 μ M HpTx1 into the hind paw of the Nav1.7-KO mice triggered robust nociceptive responses, such as licking and biting of the injected paw (Fig.1c). We further found that injection of HpTx1 increased the sensitivity to heat and mechanical stimulation in fNav1.7 and Nav1.7-KO mice, respectively (Fig.1d, e). The ability of mechanical pain sensation of Nav1.7-KO mice was recovered to normal during treatment of 10 μ M HpTx1 (Fig.1c-e).” The claims are in apparent contradiction to each other; please clarify.

DRG recordings – DRG neurons are routinely classified by diameter or area of the soma; using capacitance is rather unusual (“ <22 pF”). Please provide information on the actual size of the cells.

More importantly, only a fraction of ostensibly small neurons were HpTx1-sensitive (50 to ~65%). A full description of the cells that were insensitive needs to be provided. In addition to the values reported in Table 1, please provide information on the input resistance (R_{in}) in the absence and presence of HpTx1. Table 1 needs to include the number of cells for each condition reported.

Data in Table 1; The authors report that in WT DRG neurons RMP is -50.4 ± 1.1 mV vs. -48.3 ± 1.3 mV in the presence of 0.75 μ M HpTx1 and that the difference is significant at $p < 0.01$ (per the manuscript, errors are reported as S.E.M). The text (line 145) indicates that the sample size $n = 15$; when I run the t-test using Sigmaplot on those values, I obtain $p = 0.228$. The same issue as to accuracy of the statistics arises when comparing RMP for DRG neurons from Nav1.8 KO mice (-46.9 ± 1.9 vs 44.3 ± 1.8 ; $p = 0.329$, assuming $n = 15$ since no other value provided; if the sample size is smaller than that, it is not possible that those values will be significantly different at $p < 0.001$ as reported or at any other level).

Line 166-167 – “might enhance the membrane excitability of the small DRG neurons in WT, fNav1.7 and Nav1.7-KO mice” should be changed to read “might enhance the membrane excitability of some small DRG neurons in WT, fNav1.7 and Nav1.7-KO mice”

Line 210-211 “of 0.75 μ M HpTx1 (Control: $V_{1/2} = -69.9 \pm 1.3$ mV and $k = -9.0 \pm 0.7$ mV; HpTx1: $V_{1/2} = -56.5 \pm 2.4$ mV and $k = -15.0 \pm 0.6$ mV; $n = 8$, $P > 0.05$, the paired Student’s t-test), but Table 2 indicates that $p < 0.001$. Please reconcile.

Line 285 “binding affinity” – the authors did not measure binding affinity (often reported as KD) in any of the experiments in this manuscript . Perhaps the correct word is “efficacy”? (same basic issue on lines 174 and possibly 310).

A key set of experiments needs to be performed; as shown in Fig. 3D, there is a marked right-shift in the $V_{1/2}$ for INaV1.9 recorded in ND7/23 cells in the presence of 0.75 μ M HpTx1. Comparable recordings (\pm HpTx1) need to be performed in DRG neurons from WT, Nav1.7-KO, Nav1.8-KO, Nav1.9, and Nav1.7/1.8-KO mice. One would predict that the $V_{1/2}$ in the presence of toxin will shift according to the population of Nav channels expressed, and this needs to be tested experimentally.

With regards to the chimeric data, it so hard to see what is going on. For example:

1. With respect to 1.7/1.8, they completely ignore the finding that some of their 1.8 (insensitive) into 1.7 (inhibited) mutants actually enhance the inhibition. The authors then use a linear axis in Fig 6E to totally obscure this observation; Fig 6E should be plotted on a log axis. To complete this analysis, it is necessary to perform the experiments using the reverse chimeras/point mutations.

2. For their 1.9/1.8 experiments, the authors did construct reverse chimeras and Figure 6B certainly seems to suggest certain residue are important in the differences between 1.8 and 1.9. That said, you can have a loss of potentiation arise two ways, either you have interfered with the potentiation or the mutant channel is already fully potentiated so the effect of the toxin is obscured even though still present. One can look at this using conditions where you don't maximally activate the current, and this should be done.

Other points:

A toxin (HpTx1, this manuscript) that has been previously identified as a K channel inhibitor (κ -sparatoxin-Hv1a; Sanguinetti et al. 1997 – Ref 33) which the authors now show also has interesting effects on Nav channels. But, not only do they wait until page 8 to mention this fact (section HpTx1 is an inhibitor of Nav1.7 and an activator of Nav1.9), it is presumptuous on their part to rename the molecule; clearly this is not appropriate. And add to that they downplay the potency for Kv4.2 inhibition (“moderate-affinity”) which is disingenuous given the IC_{50} is not that much higher than that of the effects on the Navs.

Figures

Figure 2 – please provide data demonstrating recovery following washout of drug. For the sample sizes presented in the legend, are the sample sizes the same for each data point in the input-output curves? Are all the recordings from the same culture dish? If not from the same dish, same animal or multiple animals?

Extended Fig 1 – what purpose does panel A serve?

Extended Fig 1 – panel B – poorly presented - should be separate images, and since the bar graph is simply a comparison of control vs. single concentration of HPTx1, this information could be presented in the text without needing to show it graphically.

REFERENCES

Bennett DL, Clark AJ, Huang J, Waxman SG, & Dib-Hajj SD (2019). The Role of Voltage-Gated Sodium Channels in Pain Signaling. *Physiol Rev* 99: 1079-1151.

Cao L, McDonnell A, Nitzsche A, Alexandrou A, Saintot PP, Loucif AJ, et al. (2016). Pharmacological reversal of a pain phenotype in iPSC-derived sensory neurons and patients with inherited erythromelalgia. *Sci Transl Med* 8: 335ra356.

Cox JJ, Reimann F, Nicholas AK, Thornton G, Roberts E, Springell K, et al. (2006). An SCN9A channelopathy causes congenital inability to experience pain. *Nature* 444: 894-898.

Cox JJ, Sheynin J, Shorer Z, Reimann F, Nicholas AK, Zubovic L, et al. (2010). Congenital insensitivity to pain: novel SCN9A missense and in-frame deletion mutations. *Hum Mutat* 31: E1670-1686.

Klein SL, Schiebinger L, Stefanick ML, Cahill L, Danska J, de Vries GJ, et al. (2015). Opinion: Sex inclusion in basic research drives discovery. *Proc Natl Acad Sci U S A* 112: 5257-5258.

McDermott LA, Weir GA, Themistocleous AC, Segerdahl AR, Blesneac I, Baskozos G, et al. (2019). Defining the Functional Role of Nav1.7 in Human Nociception. *Neuron* 101: 905-919 e908.

Minett MS, Pereira V, Sikandar S, Matsuyama A, Lolignier S, Kanellopoulos AH, et al. (2015). Endogenous opioids contribute to insensitivity to pain in humans and mice lacking sodium channel Nav1.7. *Nat Commun* 6: 8967.

Mogil JS (2012). Sex differences in pain and pain inhibition: multiple explanations of a controversial phenomenon. *Nat Rev Neurosci* 13: 859-866.

Reviewers' comments:

Reviewer #1 (Remarks to the Author):

The authors identify a novel venom-derived peptide that works to induce mechanical allodynia and pain sensitivity by activating Nav1.9, while also interestingly inhibiting Nav1.7. The site-directed mutagenesis experiments implicating site 3 or site 4 in the differential response of Nav1.9 and Nav1.7, respectively, was very convincing. However, the significance of this work is broader than this and two-fold. First, it presents a proof of principle for a new therapeutic avenue in the treatment of congenital insensitivity to pain by rescuing painlessness via a Nav1.9-mediated mechanism. Secondly, the authors have further refined and improved knowledge of the respective contributions of Nav1.7-9 to action potential electrogenesis.

R: Thank you for taking the time to make such detailed comments on our manuscript, which is very helpful for us to improve our manuscript.

Major comments

1)The discussion and implications of these results are interesting, but could be further strengthened by discussions of the following findings and papers.

R: We thank the reviewer for this suggestion. According to the reviewer's comment, we now rewrote and rearranged the Discussion of the manuscript.

a)Rush, A.M. et al, A single sodium channel mutation produces hyper-or hypoexcitability in different types of neurons. Proc. Nat. Acad. Sci., 103: 8245-8250, 2006.

i)This paper demonstrates the principle that the effect of Nav1.7 mutations on excitability can depend on cell background

R: According to the reviewer's comment, we have added some discussion in the Discussion section as follows. "In addition, Rush et al. demonstrated that the functional consequence of depolarized RMP in part also depends on the Nav1.8 expression profile of affected neurons."

b)Rush, A.M., Waxman, S.G. PGE2 increases the tetrodotoxin-resistant Nav1.9 sodium current in mouse DRG neurons via G-proteins, Brain Research, 1023: 264-271, 2004.

i)Agents such as prostaglandins can influence Nav1.9 current

c)Rush, A.M. et al, Contactin regulates the current density and axonal expression of TTX-R but not TTX-S sodium channels in DRG Neurons, Eur. J. Neurosci., 22:39-49, 2005.

i)Endogenous molecules such as contactin can influence level of expression of Nav1.9

R: According to the reviewer's comment, we have added some discussion in the Discussion section as follows. "On the other hand, it has been found that Nav1.9 activity can be modulated by various factors. For examples, glial-cell-derived neurotrophic factor (GDNF) significantly increases Nav1.9 mRNA and current density; endogenous molecules contactin promotes the surface expression of Nav1.9; some inflammatory mediators, such

as PGE2, GTP, histamine et al., can markedly increase the current density of Nav1.9 currents and AP firing.”

2) In Nav1.8-KO mice, intraplantar injections of HpTx1 failed to affect nocifensive behavior. Additionally, there was no change in mechanical threshold or withdrawal latency. This is interesting, because 1.8 KO mice theoretically express 1.7 and 1.9 like WT mice, but WT mice showed enhanced pain behavior in response to intraplantar injection of the toxin. Based on the ND7 cell data, the toxin does not directly interact with Nav1.8. The authors conclude that Nav1.9 could “largely compensate for the absence of Nav1.7 but not Nav1.8.” Could the authors expand on this? A related but distinct explanation is that the RMP of 1.8-null DRG neurons is the most depolarized, inactivating Nav1.7 and the lack of Nav1.8 now results in significantly fewer active channels to induce action potential electrogenesis. This hypothesis is based on the findings of the paper cited in point 1a, above.

R: We thank the reviewer for pointing this out, and we appreciate the comment. Regarding HpTx-I is ineffective on Nav1.8 KO mice, we explained as follows: because Nav1.9 does not contribute much to the amplitude of APs, so it is suggested that the enhanced Nav1.9 activity could not compensate for the loss of Nav1.8 which contributes to the majority (58-90%) of the inward current during the rising phase of an all-or-none AP in nociceptive sensory neurons, consistent with that HpTx1 failed to affect the AP firing in the Nav1.8-KO DRG neurons and the pain sensitivity in the Nav1.8-KO mice. In addition, Rush et al. demonstrated that the functional consequence of depolarized RMP in part also depends on the Nav1.8 expression profile of affected neurons. Therefore, another possible explanation is that depolarization of the RMP of DRG neurons lacking Nav1.8 might result in inactivation of some other channels and therefore significantly fewer active channels to induce AP electrogenesis.

The above discussion has been added in the Discussion section in the reviewed manuscript.

3) When neurons are lacking Nav1.9, the effect of the toxin is clearly to induce analgesia (increased current threshold, decreased AP firing, increased mechanical threshold and latency of paw withdrawal). When neurons are lacking Nav1.7, the effect of the toxin is clearly to induce pain-sensitivity (reduced mechanical threshold and reduced latency to paw withdrawal). WT mice, however, presumably expressing both Nav1.7 and -1.9 exhibited reduced threshold and latency. This seems to indicate that the effect on Nav1.9 predominates over the effect on Nav1.7. Could you hypothesize as to why this is?

R: This is an important comment. We think this is related with the roles and the relationship of the two channels (Nav1.7 and Nav1.9) in AP generation. We discussed this as follows: the WT mice which express Nav1.7 and Nav1.9 just displayed pain responses with the treatment of HpTx1, indicating enhanced Nav1.9 activity might conceal the analgesic activity derived from the inhibition of Nav1.7. This was possibly conceivable, if we considered inhibition of Nav1.7 was analogous to Nav1.7 knock out to some extent. Based on the role of Nav1.7 or Nav1.9 as a threshold channel for AP, a possible reason is that HpTx1-bound Nav1.9 largely depolarized RMP of DRG neurons, which could compensate for the absence of Nav1.7 and might be sufficient for AP generation. Studies

have proved that a moderately depolarized RMP contributes to the decrease of current threshold for AP generation by promoting other ion channels open (i.e., Nav1.8 and voltage-gated calcium channels). Our data demonstrate that HpTx1 depolarize RMP by 2.0 mV and 3.2 mV in the WT and Nav1.7 KO mice, respectively, which leads to reduced current threshold and increased AP and finally enhances the membrane excitability of some small DRG neurons. In the normal condition, since the low amount of Nav1.9 expressed in DRG neurons, its contribution in the AP firing of pain signaling may be not prominent. Compared with Nav1.7 or Nav1.8, no loss-of-function mutants have been found in clinical and the Nav1.9-KO mice do not show evident change of physiological nociception. On the other hand, in the hyperalgesia conditions, for example, with the treatment of HpTx1, inflammatory factor, cold and gain-of-function mutations, the contribution of Nav1.9 can be upgraded and even compensate the loss of Nav1.7. Therefore, the enhanced activation of Nav1.9 can recover the pain responses in Nav1.7-related CIP as revealed in our studies.

Minor comments

1) Only 50% of WT small-diameter DRG neurons were HpTx1-sensitive (reduced current threshold, increased AP firing, etc.). Not all DRG neurons express Nav1.9 at high or appreciable levels. It would be interesting to note whether IB4-positivity was associated with the 50% of DRG neurons that responded to the toxin versus a lack of IB4-positive staining in nonresponsive cells. If no staining was conducted, could you tell, based off sample traces, if Nav1.9 currents were present in these non-responder neurons (assuming voltage-clamp was also done on these cells).

R: We thank the reviewer for this comment. We added IB4 staining experiment in the reviewed manuscript, and observed a strong overlap (59.6%) between Nav1.9 and IB4 (Extended Data Fig.4), which appeared to be consistent with the expression of Nav1.9 in small-diameter DRG neurons.

2) Building on that point, was a subgroup analysis conducted of the HpTx1 non-responding neurons, comparing them to WT neurons injected with control? If the non-responders had minimal Nav1.9 currents, one would expect that these neurons might show an increased rheobase compared to WT neurons that were injected with control, especially since Nav1.9-KO neurons had an increase rheobase when exposed to the toxin

R: We thank the reviewer for this suggestion. According to the reviewer's comment, we have analyzed the change (decreased, unchanged and increased) of current threshold of six kinds of mice (WT, fNav1.7, Nav1.7-KO, Nav1.8-KO, Nav1.9-KO, Nav1.7/Nav1.8-DKO) DRG neurons after HpTx1 treatment. Specifically, Table 3 has been added to the reviewed manuscript (Table 3). The WT, fNav1.7, Nav1.7-KO, and Nav1.7/Nav1.8-KO DRG neurons showed a higher proportion with decreased current threshold than Nav1.9-KO DRG neurons when exposed to the toxin. Not only that, the current thresholds of 48% Nav1.9-KO DRG neurons were increased.

3) Nav1.6 is also found within DRG neurons. In WT mouse DRG neurons, you found that 62.9% of TTX-S currents were inhibited by 1 micromolar HpTx-1. Was the effect of the

toxin on Nav1.6 investigated? The data showing the inhibitory effect on Nav1.7 is strong on its own, but a small discussion on the Vasylyev et al 2014, which showed that roughly 70% of TTX-S current in mice, would lend further evidence to this reduction in TTX-S current being mediated by Nav1.7 and not Nav1.6.

R: We thank the reviewer for this comment. Actually, HpTx1 had inhibitory activity on Nav1.6 currents and the IC_{50} value was determined as $5.63 \pm 0.13 \mu\text{M}$, which was 10-fold less potent than that of on Nav1.7 currents. Vasylyev et al estimated that ~70% TTX-S Nav currents are mediated by Nav1.7 in small DRG neurons. These data suggested that the observed effect on TTX-S Nav currents should mainly arise from Nav1.7 inhibition by HpTx1.

REFERENCES

1. Vasylyev DV, Han C, Zhao P, Dib-Hajj S, Waxman SG. Dynamic-clamp analysis of wild-type human Nav1.7 and erythromelalgia mutant channel L858H. *J Neurophysiol.* 2014 :111(7):1429-43.

4) In figure 2a, the HpTx1-induced calcium release looks significantly closer to background (almost unchanged, even) and significantly less like the positive-control. Figure 2b shows the average ratiometric calcium response increasing from 1 to approximately 1.05 after application. Is this significant?

R: Yes, this is significant. Thanks for you pointing this out. As shown in Fig. R1, we conducted this experiment again and counted more cell. As a result, a total of 138 DRG neurons were recorded. We still found that HpTx1 triggered little but significant change in calcium response (Control: 0.005 ± 0.01 ; $0.75 \mu\text{M}$ HpTx1: 0.15 ± 0.02 , **** $P < 0.0001$, Mann-Whitney test). In order to avoid misunderstanding, we have removed this experiment in the reviewed manuscript. Indeed, this experiment is not completely necessary to our findings, and the results of the current-clamp recordings are sufficient to prove that HpTx1 enhances the excitability of DRG neurons.

Figure.R1 HpTx1 triggers calcium responses in DRG neurons. (a) $0.75 \mu\text{M}$ HpTx1 triggers calcium responses in acutely dissociated mouse DRG neurons which are loaded with the calcium indicator Fluo-4 AM. High K^+ ($60 \text{ mM } K^+$) acts as positive control. Scale bar, $30 \mu\text{m}$. (b) Average ratiometric calcium responses from total (black, $n=138$), response (blue, $n=71$) or no-response (red, $n=67$) DRG neurons.

Reviewer #2 (Remarks to the Author):

In this study entitled “Activating Nav1.9 recovers pain responses in Nav1.7-related CIP as revealed by a spider toxin”, Zhou et al characterize the effects of HpTx1 on different types of sodium channels and, by using various Nav channel KO mice, characterize the effects of HpTX1 on pain behavior and DRG neuron excitability. There is a lot of work presented, and some of the results are interesting, but I have many reservations.

R: Thank you for your time in reviewing our manuscript, and thank you for the comments, which are very helpful for us to improve our manuscript.

Major concerns.

1-The premise that congenital insensitivity to pain (CIP) could be treated using a toxin such as HpTX1 seems far-fetched. Leaving aside that not all forms of CIP are due to loss-of-function mutations of Nav1.7, this toxin causes spontaneous pain in addition to restoring sensitivity to painful stimuli. The latter effect is good (assuming it could be achieved in a sustained way without causing adverse effects... including via blockade of Kv4.2) but the former is obviously very bad. The results of the study are interesting without trying to find some clinical application that isn't well thought out.

R: We thank the reviewer for pointing this out. Actually, we completely consent the comment. As the reviewer considered, to date, some gene mutations have been found to cause congenital insensitivity to pain (CIP). For examples, mutations in NGF or *NTRK* (which encodes the NGF high-affinity receptor) result in cell death and axonal degeneration of small-diameter sensory neurons^{1,2}, and mutations in PRDM12 (which belongs to a family of histone methyltransferases, and with a crucial role to vertebrate neurogenesis) cause a loss of small-calibre myelinated axons³. In contrast, loss of function mutations in Nav1.7 have a structurally normal peripheral nervous system, which causes CIP because of impaired electrical signaling function rather than impaired structural integrity of nociceptors⁴. Based on our results, HpTx1 can enhance Nav1.7-KO DRG neurons activity to trigger pain in Nav1.7-KO mice, suggesting that HpTx1 may be useful to recover the pain responses of Nav1.7-related CIP. It is greatly possible this HpTx1 should have no efficacy on Nav1.7-unrelated CIP. However, we think this would not reduce the importance of our study.

Considering the relatively low selectivity of HpTx1 to Nav1.9, it might possibly bring about some side effects, and it is necessary to improve the selectivity of HpTx1 in the further study. But importantly, this study provided a new insight into the treatment of Nav1.7-related CIP by activating Nav1.9, and that provides a direction for drug development. For example, it has been found that some endogenous molecules can increase the current density of Nav1.9, therefore the strategy enhancing activation Nav1.9 by regulation of these endogenous molecules may be a feasible strategy for treating CIP.

REFERENCES

1. Einarsdottir E, Carlsson A, Minde J, et al. A mutation in the nerve growth factor beta gene (NGFB) causes loss of pain perception. *Hum Mol Genet* 2004; 13: 799–805.
2. Indo Y, Tsuruta M, Hayashida Y, et al. Mutations in the TRKA/NGF receptor gene in patients with congenital insensitivity to pain with anhidrosis. *Nat Genet* 1996; 13: 485–88.

3. Chen YC, Auer-Grumbach M, Matsukawa S, et al. Transcriptional regulator PRDM12 is essential for human pain perception. *Nature genetics* 2015; 47(7):803-8.
4. Cox JJ, Reimann F, Nicholas AK, et al. An SCN9A channelopathy causes congenital inability to experience pain. *Nature* 2006; 444: 894–98.

2-The presentation of data and statistical analysis raise some significant concerns. In many cases the sample size is not reported, and when it is mentioned in a figure legend, it is often unclear which panel it refers to, and there are inconsistencies with the main text. There is no mention of whether the data are normally distributed and, therefore, whether parametric tests are appropriate. Individual data points are not shown on the majority of graphs, and so the reader cannot even start to judge this. Many of the answers provided on the Reporting Summary are demonstrably wrong.

R: We thank the reviewer for pointing this out. According to the suggestion, we performed more appropriate statistical analyses of the whole data in the revised manuscripts, and we now mentioned in the figure legend the sample size and the statistical tests used for each analysis. We clarified the statistics in the Method section and the Reporting Summary. All values are shown as mean \pm S.E.M and n represents the number of animals or cells examined. At least six or more mice were used in animal experiment to obtain statistical analysis. Experiments were performed at least three independent times. Sample size in all experiments were estimated based in previous published experiments. Assessment of normality for sample ≥ 11 was tested using the Kolmogorov-Smirnov normality test. When the data are normally distributed, the parametric paired or unpaired t-test was analyzed. If the data are not normally distributed, we typically used non-parametric Mann-Whitney test, Wilcoxon matched-pairs signed rank test to test. Small sample size lacks power to test normality, therefore we typically used the unpaired and paired Student's two tail t-test, or one-way ANOVA with Tukey's /Dunnett's Multiple Comparison Test to test. Figure legends specify which test was used for specific experiments. Significant levels were set at $P < 0.05$. Statistical analyses were performed with Prism 7.0 (GraphPad) softwares.

3-In Figure 2b, the graph shows a single curve which is apparently the average response for 20 neurons. The curves for individual cells are not shown, nor are there any standard error markings around the group curve. Furthermore, I'm wondering if the authors have included all cells, or just cells that responded to the HpTX (this comes up again in my next point). 20 cells is very little for a calcium imaging experiment. Lastly, the figure legend refers to Fura-2, but the Methods discuss Fluo-2. Which is it?

R: Our mistake. Fura-2AM was used.

As shown in Fig. R1, we conducted this experiment again and counted more cell. As a result, a total of 138 DRG neurons were recorded. We still found that HpTx1 triggered little but significant change in calcium response (Control: 0.005 ± 0.01 ; $0.75 \mu\text{M}$ HpTx1: 0.15 ± 0.02 , **** $P < 0.0001$, Mann-Whitney test). In order to avoid misunderstanding, we have removed this experiment in the reviewed manuscript. Indeed, this experiment is not completely necessary to our findings, and the results of the current-clamp recordings are sufficient to prove that HpTx1 enhances the excitability of DRG neurons.

Figure.R1 HpTx1 triggers calcium responses in DRG neurons. (a) 0.75 μM HpTx1 triggers calcium responses in acutely dissociated mouse DRG neurons which are loaded with the calcium indicator Fluo-4 AM. High K^+ (60 mM K^+) acts as positive control. Scale bar, 30 μm . (b) Average ratiometric calcium responses from total (black, $n=138$), response (blue, $n=71$) or no-response (red, $n=67$) DRG neurons.

4-On p 7, the authors explain that 15 out of 30 DRG neurons exhibited a reduced current threshold after HpTX. In Fig. 2c-f, does that mean that only data from HpTX-“responsive” cells are being reported. The figure legend does not provide the necessary information to resolve this confusion. If the authors have done what I think they’ve done, it’s totally unacceptable to remove data from cells simply because they don’t respond, and then claim that the responsive cells show a significant response. A change in the distribution of the data (because of mixture of responsive and unresponsive cells) could be shown statistically (e.g. Kolmogorov-Smirnov test) and it would be worth trying to identify an independent criterion by which identify responsive vs unresponsive cells (e.g. peptidergic vs. non-peptidergic small DRG cells), but what the authors seem to have done is unacceptable. The same problem seems to apply to the latter half of Figure 2.

R: We agree this is an important question that needs to be clarified, thanks! Indeed, we did not remove the HpTx1 no-responsive cell data during data analysis. In our current-clamp recording experiments, we detect the change of RMP, current threshold and AP amplitude of the small DRG neurons after treatment with 0.75 μM HpTx1. Each recording data is derived from before and after treatment 0.75 μM HpTx1 in the same of neurons. In the revised manuscript, we redrew these figures and showed all data derived from responsive and unresponsive cells, which clearly showed that changes of individual cells. And Table 3 was added, which displayed the change (decreased, unchanged and increased) of current threshold of DRG neurons from six kinds of mice (WT, fNav1.7, Nav1.7-KO, Nav1.8-KO, Nav1.9-KO, Nav1.7/Nav1.8-DKO) after HpTx1 treatment.

5-I think it is misleading when the authors write that the “identified” a spider peptide toxin called HpTX1. It is not until halfway through the Results (p 8) that the authors mention that this toxin was previously identified and is known to block certain potassium channels.

R: We feel sorry for this mistake and appreciate your comment. We have solved this problem in the new manuscript.

6-The author refer to extended figure 1e and f as evidence that HpTx1 does not have any effect on DRG potassium or calcium channels. But those panels only show a single

sample trace showing no effect of the toxin. That is completely insufficient to show a lack of effect.

R: Thanks for this suggestion. We did follow your advice and added scatter plot (Extended Data Fig.1c and d) to show the remaining currents after treatment of HpTx1.

7-Were the behavioral tests blinded? This is mentioned in the reporting summary, but not in the manuscript. Were any of the electrophysiological tests blinded?

R: We thank you for this comment. The behavioral tests were blinded. We now mention in the section of "Material and methods". Electrophysiological recordings did not blind.

8-I am struck by the number of times review articles are cited when referring to specific findings. For instance, at the bottom of p 3, the authors explain the contribution of Nav1.8 and Nav1.7 to different parts of the action potential, each time citing a different review article. If this is a key issue – which I think it is – then the primary studies should be cited.

R: We feel sorry for this mistake and appreciate your comment. We have solved this problem in the reviewed manuscript.

9-Was the liquid junction potential accounted for when reporting the data. On p 24, it was mentioned that liquid junction potential was zeroed before seal formation, but that does not remove the effect after breaking in and recording in whole-cell mode.

R: This is an important comment. In our experiments, the data were obtained in the Axopatch 200B series amplifier, and the liquid junction was corrected. However, when the whole-cell recording performed in the EPC-10 USB amplifier, the liquid junction potential was not corrected, because it is less than 3 mV and has little effect on the recording. We now mentioned in the section of "Material and methods".

Other concerns:

-For Fig 3d, were all of those experiments conducted in TTX.

R: Yes, all of TTX-R Nav channels recording experiments were conducted in 1 μ M TTX, we now mentioned in the figure legend ("Note that 1 μ M TTX were applied in those experiments").

-*s are referred to later in the figure 3 legend, but none are shown on the figure.

R: Thanks! We have corrected.

-Extended data figure 3 refers to n=4, but only sample traces are shown on the figure.

R: Thanks for this suggestion. We have added scatter plot in the reviewed manuscript (Extended Data Fig.3).

Reviewer #3 (Remarks to the Author):

Main text is 6,459 words (Intro, Methods, Results, Discussion) – this needs to be considerably shortened (Per instructions to Authors, “main text of no more than 5,000 words”). The Introduction rambles and needs to be written and significantly shortened.

R: Thank you for taking the time to make such detailed comments on our manuscript, which are very helpful for us to improve our manuscript. We have rewritten the manuscript, and it can meet the journal's standard.

The English writing (grammar, noun-verb agreement, word choice, claims/statements of fact, etc.) is not acceptable in the current submission; it needs to be carefully reviewed and edited by a native speaker of English; examples include, but are by no means limited to:

Line 17 “nociceptor” should be “nociceptors”

Line 19 “loss” should be “lose”

Line 20-22 “So far, the therapeutic agents for treating the Nav1.7-related CIP are still in its infancy” – as written, this implies that such therapeutic agents exist (Minett et al., 2015); but this is possibly the only example, and caution should be used when citing it.

Line 24 “conformed” should be “confirmed”

R: We thank you for your thorough reading of the manuscript. We have corrected in the reviewed manuscript, and the manuscript has been thoroughly revised and edited by a native speaker.

Line 33 “and propose that Nav1.7, Nav1.8 and Nav1.9 are interrelated in pain signaling” – this “proposal” has long been put forth by Steve Waxman and his colleagues for over two decades – see recent review (Bennett, Clark, Huang, Waxman & Dib-Hajj, 2019).

R: Thanks for your suggestion. We have improved the manuscript wording throughout as well as we were much more careful in some conclusion. Here, this sentence have replaced by “and offer a pharmacological insight into the relationship of the three Nav channels in pain signaling”.

And that is just in the Abstract; there are comparable problems throughout. It is the responsibility of the authors to have the manuscript in its entirely reviewed and revised appropriately.

R: We are very sorry for the mistakes in this manuscript. The revised manuscript has been edited by a native speaker, so we hope it can meet the journal's standard. Thanks for your useful comments.

Introduction:

Lines: 54-58 – clinical trials with NaV1.7 blockers – to merely state that trials have been undertaken is to minimize the point the authors are trying to make – the molecules actually show efficacy, but that point needs to be carefully conveyed (the study by (Cao et al., 2016), for example, only examined five subjects, and the effect was modest).

R: We thank the reviewer for these suggestions and implemented these changes. We now rewrote this section, and the sentence have replaced by “Sustained efforts have been making to develop selective inhibitors of this channel and some have shown efficacy in clinical studies, although larger clinical trials are needed to definitely access efficacy”

Line 63 “Urgent” – NaV1.7 loss-of-function mutations leading to congenital insensitivity to pain are extremely rare (Cox et al., 2006; Cox et al., 2010; McDermott et al., 2019); to say that a treatment is “urgently” needed is overstating the case. “needed” will suffice.

R: Thanks for your suggestion. We have improved the manuscript wording throughout.

Lines 62-63, References 22,23,24 – discussion of animal venoms – all the cited references are to specific NaV1.1 blockers. Surely there are others and they should be cited.

R: We thank the reviewer for this suggestion. We have added other references in the revised manuscript.

Line 75 – “It produces a large window current” – this is not a “stand alone” sentence.

R: We thank the reviewer for making us aware of this mistake. We rewrote and rearranged the Introduction section of the manuscript, and this sentence was removed.

Methods/Results

Mice – “6-8 week-old adult C57Bl6” – at best these would be categorized as “young adult” but to avoid a semantic discussion, simply remove the word “adult”.

R: Thanks, we have corrected.

The authors need to specify throughout the manuscript whether behavioral studies were sex (male:female)-balanced, and need to indicate sex of the mice used to provide DRG neurons. If only a single sex was used in either or both of these experiments, this will present significant problems with respect to understanding the significance of the observations given that profound sex-based differences are present when evaluating pain-related behaviors and their underlying mechanisms (Klein et al., 2015; Mogil, 2012).

R: We agree this is an important question that needs to be clarified. Actually, in the study, we considered the effect of sex on animal behavioral tests. The behavioral tests were sex-balanced (male: female =1:1), but the DRG neurons were from the mixed of one male and one female mouse. We now mentioned in the section of “Material and methods”.

Results

Line 104-105; “The venom fractions were screened for pain-induced activity.” – 15 crude venoms were fractionated – please indicate how many purified samples were obtained

following fractionation. Of that number please confirm that all other purified fractions failed to elicit nocifensive behaviors.

R: We thank the reviewer for pointing this out. We have added some information in the Results section: “We first fractionated 15 crude venoms (from 10 spiders and 5 snakes) using semipreparative reversed-phase high performance liquid chromatography (RP-HPLC), and a total of 110 fractions (5-10 fractions for per venom) were collected. The venom fractions were screened for pain-induced activity, and 6 fractions were positive, in which a fraction in the venom from the spider *H. venatoria* exhibited the strongest effect (Fig.1a). We further purified this fraction by analytical RP-HPLC and found a component having such an effect.”

Lines 123-127 – “Similar to control (fNav1.7) littermate mice, injection of 10 μ M HpTx1 into the hind paw of the Nav1.7-KO mice triggered robust nociceptive responses, such as licking and biting of the injected paw (Fig.1c). We further found that injection of HpTx1 increased the sensitivity to heat and mechanical stimulation in fNav1.7 and Nav1.7-KO mice, respectively (Fig.1d, e). The ability of mechanical pain sensation of Nav1.7-KO mice was recovered to normal during treatment of 10 μ M HpTx1 (Fig.1c-e).” The claims are in apparent contradiction to each other; please clarify.

R: We apologize for the ambiguity. We rewrote this part and now this is stated as “injection of 10 μ M HpTx1 into the hind paw of the Nav1.7-KO mice or the control (fNav1.7) littermate mice triggered robust nocifensive behaviors, such as licking and biting of the injected paws. Furthermore, the pain-inducing effect was further validated in evoked pain models (Fig.1d, e). The Nav1.7-KO mice with the treatment of 10 μ M HpTx1 recovered the deficit in mechanical pain caused by Nav1.7 ablation (Fig.1d); injection of 10 μ M HpTx1 also reduced thresholds for thermal pain in the Nav1.7-KO mice, paralleling the effect of HpTx1 on the fNav1.7 mice (Fig.1e).”

DRG recordings – DRG neurons are routinely classified by diameter or area of the soma; using capacitance is rather unusual (“<22 pF”). Please provide information on the actual size of the cells.

R: We thank reviewers for pointing this out. We have corrected in new manuscript.

More importantly, only a fraction of ostensibly small neurons were HpTx1-sensitive (50 to ~65%). A full description of the cells that were insensitive needs to be provided. In addition to the values reported in Table 1, please provide information on the input resistance (R_{in}) in the absence and presence of HpTx1. Table 1 needs to include the number of cells for each condition reported.

R: Thanks for your suggestion. We have analyzed the changed (decreased, unchanged and increased) current threshold of DRG neurons from six kinds of mice (WT, fNav1.7, Nav1.7-KO, Nav1.8-KO, Nav1.9-KO, Nav1.7/Nav1.8-DKO) after HpTx1 treatment. Specifically, Table3 has been added to the manuscript (Table3), and the input resistance

and sample size have been added in the eTable1.

Data in Table 1; The authors report that in WT DRG neurons RMP is -50.4 ± 1.1 mV vs. -48.3 ± 1.3 mV in the presence of 0.75 μ M HpTx1 and that the difference is significant at $p < 0.01$ (per the manuscript, errors are reported as S.E.M). The text (line 145) indicates that the sample size $n = 15$; when I run the t-test using Sigmaplot on those values, I obtain $p = 0.228$. The same issue as to accuracy of the statistics arises when comparing RMP for DRG neurons from Nav1.8 KO mice (-46.9 ± 1.9 vs 44.3 ± 1.8 ; $p = 0.329$, assuming $n = 15$ since no other value provided; if the sample size is smaller than that, it is not possible that those values will be significantly different at $p < 0.001$ as reported or at any other level).

R: We thank you for noticing it. In our current-clamp recording experiments, we detected the change of RMP of the small DRG neurons after treatment with 0.75 μ M HpTx1. This recording is derived from the same cell. So, the paired t-test was used during data statistics. Following the reviewer's comment, we have now carefully revised the statistical tests used for each figure. For the WT and Nav1.8-KO DRG neurons, RMP data were from 30 and 18 independent cells, respectively. these data were normally distributed and analyzed with a parametric paired t-test (WT mice: Control= -50.4 ± 1.2 mV, 0.75 μ M HpTx1= -48.4 ± 1.3 mV, $n=30$, $***P<0.001$, the parametric paired t-test; Nav1.8-KO mice: Control= -46.9 ± 1.9 mV, 0.75 μ M HpTx1= -44.3 ± 1.8 mV, $n=18$, $**P<0.01$, the parametric paired t-test). We now mentioned these in the figure legend the statistical test used for each analysis. Meanwhile, Source data are provided as a Source Data file.

Line 166-167 – “might enhance the membrane excitability of the small DRG neurons in WT, fNav1.7 and Nav1.7-KO mice” should be changed to read “might enhance the membrane excitability of some small DRG neurons in WT, fNav1.7 and Nav1.7-KO mice”

R: Thanks for noticing that. We have corrected in the reviewed manuscript.

Line 210-211 “of 0.75 μ M HpTx1 (Control: $V_{1/2} = -69.9 \pm 1.3$ mV and $k = -9.0 \pm 0.7$ mV; HpTx1: $V_{1/2} = -56.5 \pm 2.4$ mV and $k = -15.0 \pm 0.6$ mV; $n = 8$, $P>0.05$, the paired Student's t-test), but Table 2 indicates that $p < 0.001$. Please reconcile.

R: Our mistake, we thank you for pointing this out. It should be $P<0.001$.

Line 285 “binding affinity” – the authors did not measure binding affinity (often reported as KD) in any of the experiments in this manuscript . Perhaps the correct word is “efficacy”? (same basic issue on lines 174 and possibly 310).

R: Thanks for noticing that. We have corrected in the reviewed manuscript.

A key set of experiments needs to be performed; as shown in Fig. 3D, there is a marked right-shift in the $V_{1/2}$ for INaV1.9 recorded in ND7/23 cells in the presence of 0.75 μ M HpTx1. Comparable recordings (\pm HpTx1) need to be performed in DRG neurons from WT, Nav1.7-KO, Nav1.8-KO, Nav1.9, and Nav1.7/1.8-KO mice. One would predict that the

V_{1/2} in the presence of toxin will shift according to the population of NaV channels expressed, and this needs to be tested experimentally.

R: We thank the reviewer for this comment. According to reviewer advices, we have added a series of experiments in the reviewed manuscript. We found that HpTx1 does not affect the TTX-R (Nav1.8) currents of Nav1.9-KO DRG neurons, which is consistent with the observation that HpTx1 did not affect the rNav1.8 currents expressed in ND7/23 cells. However, similar to the effect of HpTx1 on hNav1.9, HpTx1 significantly slowed the fast inactivation of TTX-R channels of DRG neurons in WT, Nav1.7-KO, Nav1.8-KO and Nav1.7/Nav1.8-DKO mice, and a steady-state component (approximately 35%-42% of the transient peak inward current) that was resistant to inactivation was observed in the steady-state inactivation curve (Fig.3f and Extended Data Fig.2b). Unexpectedly, HpTx1 did not alter the V_{1/2} (inactivation) of DRG TTX-R channels, which were greatly different from the effect of HpTx1 on hNav1.9 expressed in ND7/23 cells (Extended Data Fig.2b). The possible reasons for the differences were included in the Results section in the revised manuscript. "This was different from the effect of HpTx1 on TTX-R Nav currents in DRG neurons, which might be explained by their different cell types and their sequence differences. Nav1.9 channels in ND7/23 cells and DRG neurons have distinct posttranslational modification¹ and auxiliary subunits which are known to affect the pharmacological properties² and physiological properties of Nav channels."

REFERENCES

1. Bosmans F, Milescu M, Swartz KJ. Palmitoylation influences the function and pharmacology of sodium channels. *Proc Natl Acad Sci USA* 2011;108, 20213–20218
2. Zhang MM, Wilson MJ, Gajewiak J, et al. Co-expression of Nav β subunits alters the kinetics of inhibition of voltage-gated sodium channels by pore-blocking μ -conotoxins. *British Journal of Pharmacology* 2013; 168, 1597–1610.

With regards to the chimeric data, it so hard to see what is going on. For example:

1. With respect to 1.7/1.8, they completely ignore the finding that some of their 1.8 (insensitive) into 1.7 (inhibited) mutants actually enhance the inhibition. The authors then use a linear axis in Fig 6E to totally obscure this observation; Fig 6E should be plotted on a log axis. To complete this analysis, it is necessary to perform the experiments using the reverse chimeras/point mutations.

R: We thank the reviewer for this comment. We revisited the previous data. The two mutants of Nav1.7 (F813S and G819S) really increased 3-fold efficacy for the toxin (Fig. 6e, f). Furthermore, we constructed the reverse chimaeras (Nav1.8/Nav1.7 DII s3b-s4) which conferred toxin sensitivity, and the IC₅₀ value is 3.0 ± 0.6 μ M (Fig. 6e). But, reverse site-directed mutants were toxin-insensitive (Extended Data Fig. 6). These results suggested that the toxin interaction with Nav1.7 involves multiple sites. These new results were added in the revised manuscript.

2. For their 1.9/1.8 experiments, the authors did construct reverse chimeras and Figure 6B

certainly seems to suggest certain residue are important in the differences between 1.8 and 1.9. That said, you can have a loss of potentiation arise two ways, either you have interfered with the potentiation or the mutant channel is already fully potentiated so the effect of the toxin is obscured even though still present. One can look at this using conditions where you don't maximally activate the current, and this should be done.

R: We completely agree with your comment. To address the comment, the current-voltage (I–V) relationships of the mutants before and after the application of HpTx1 were explored. Three mutants (T1444L, E1450L and N1451K) significantly reduced the toxin activity at all voltages (Extended Data Fig.5f-h). Indeed, in this study, except for enhanced current amplitude, the slowed fast inactivation was considered simultaneously. HpTx1 failed to affect the fast inactivation of three mutants (T1444L, E1450L and N1451K) (Fig. 6c). Therefore, we suggested that these residues might directly affect Nav1.9 interacting with HpTx1.

Other points:

A toxin (HpTx1, this manuscript) that has been previously identified as a K channel inhibitor (κ -sparatoxin-Hv1a; Sanguinetti et al. 1997 – Ref 33) which the authors now show also has interesting effects on Nav channels. But, not only do they wait until page 8 to mention this fact (section HpTx1 is an inhibitor of Nav1.7 and an activator of Nav1.9), it is presumptuous on their part to rename the molecule; clearly this is not appropriate.

R: We feel sorry for this mistake and appreciate your comment. Indeed, HpTx1 is a short name that was named by Sanguinetti et al, while κ -sparatoxin-Hv1a is a rational nomenclature for naming peptide toxins. We rewrote and rearranged the manuscript, and mentioned in the Results section (The component was named as HpTx1 (rational nomenclature as κ -sparatoxin-Hv1a) which was first identified by Sanguinetti et al.)

And add to that they downplay the potency for Kv4.2 inhibition (“moderate-affinity”) which is disingenuous given the IC₅₀ is not that much higher than that of the effects on the Navs.

R: Thanks for your suggestion. We rewrote this part and now this is stated as “Based on Sanguinetti et al. study, HpTx1 is a blocker of Kv4.2 voltage-gated potassium (Kv) channel. Our study validated the inhibition of HpTx1 on Kv4.2 currents, and its IC₅₀ value was measured as $1.2 \pm 0.3 \mu\text{M}$ ”.

Figures

Figure 2 – please provide data demonstrating recovery following washout of drug. For the sample sizes presented in the legend, are the sample sizes the same for each data point in the input-output curves? Are all the recordings from the same culture dish? If not from the same dish, same animal or multiple animals?

R: We thank the reviewer for pointing this out. Indeed, in our another study, we found that the time course for $0.75 \mu\text{M}$ HpTx1 inhibiting the fast inactivation of Nav1.9 was characterized by a slow onset of action and an irreversible upon washing (Figure.R2).

As shown in Fig. R1, we conducted this experiment again and counted more cell. As

a result, a total of 138 DRG neurons were recorded. We still found that HpTx1 triggered little but significant change in calcium response (Control: 0.005 ± 0.01 ; $0.75 \mu\text{M}$ HpTx1: 0.15 ± 0.02 , **** $P < 0.0001$, Mann-Whitney test). In order to avoid misunderstanding, we have removed this experiment in the reviewed manuscript. Indeed, this experiment is not completely necessary to our findings, and the results of the current-clamp recordings are sufficient to prove that HpTx1 enhances the excitability of DRG neurons.

Figure.R2 Time course for the enhancement of peak currents of Nav1.9 by $0.75 \mu\text{M}$ HpTx1 and the recovery upon washing with bath solution.

Figure.R1 HpTx1 triggers calcium responses in DRG neurons. (a) $0.75 \mu\text{M}$ HpTx1 triggers calcium responses in acutely dissociated mouse DRG neurons which are loaded with the calcium indicator Fluo-4 AM. High K^+ (60 mM K^+) acts as positive control. Scale bar, $30 \mu\text{m}$. (b) Average ratiometric calcium responses from total (black, $n=138$), response (blue, $n=71$) or no-response (red, $n=67$) DRG neurons.

Extended Fig 1 – what purpose does panel A serve?

R: According to the suggestion, we have removed the panel a (Extended data Fig. 1) in the revised manuscript.

Extended Fig 1 – panel B – poorly presented - should be separate images, and since the bar graph is simply a comparison of control vs. single concentration of HPtx1, this information could be presented in the text without needing to show it graphically.

R: According to the suggestion, we have removed the graph in the revised manuscript.

REFERENCES

Bennett DL, Clark AJ, Huang J, Waxman SG, & Dib-Hajj SD (2019). The Role of Voltage-Gated Sodium Channels in Pain Signaling. *Physiol Rev* 99: 1079-1151.

Cao L, McDonnell A, Nitzsche A, Alexandrou A, Saintot PP, Loucif AJ, et al. (2016). Pharmacological reversal of a pain phenotype in iPSC-derived sensory neurons and patients with inherited erythromelalgia. *Sci Transl Med* 8: 335ra356.

Cox JJ, Reimann F, Nicholas AK, Thornton G, Roberts E, Springell K, et al. (2006). An SCN9A channelopathy causes congenital inability to experience pain. *Nature* 444: 894-898.

Cox JJ, Sheynin J, Shorer Z, Reimann F, Nicholas AK, Zubovic L, et al. (2010). Congenital insensitivity to pain: novel SCN9A missense and in-frame deletion mutations. *Hum Mutat* 31: E1670-1686.

Klein SL, Schiebinger L, Stefanick ML, Cahill L, Danska J, de Vries GJ, et al. (2015). Opinion: Sex inclusion in basic research drives discovery. *Proc Natl Acad Sci U S A* 112: 5257-5258.

McDermott LA, Weir GA, Themistocleous AC, Segerdahl AR, Blesneac I, Baskozos G, et al. (2019). Defining the Functional Role of Nav1.7 in Human Nociception. *Neuron* 101: 905-919 e908.

Minett MS, Pereira V, Sikandar S, Matsuyama A, Lolignier S, Kanellopoulos AH, et al. (2015). Endogenous opioids contribute to insensitivity to pain in humans and mice lacking sodium channel Nav1.7. *Nat Commun* 6: 8967.

Mogil JS (2012). Sex differences in pain and pain inhibition: multiple explanations of a controversial phenomenon. *Nat Rev Neurosci* 13: 859-866.

Reviewers' comments:

Reviewer #1 (Remarks to the Author):

the revisions have made this paper stronger. now acceptable in my view.

Reviewer #2 (Remarks to the Author):

Most of my original concerns have not been adequately addressed by the revisions. There is a huge amount of work presented in this study but critical gaps remain. As a result, the interpretation of the data is still quite speculative, in my opinion. The numbered points correspond to the points in my original review.

1. My first major comment about whether this toxin (or some other more specific manipulation of Nav1.9) is a viable treatment for CIP remains. The authors have added wording to specify Nav1.7-related CIP but I still think that this is a big stretch. Notably, Minett et al. (Nature Comm 2015) showed that endogenous opioids are upregulated after Nav1.7 deletion. This paper is not cited. According to it, naloxone could help restore pain perception in Nav1.7-null mice. If the author wish to emphasize that treatment of Nav1.7-related CIP, then the compensatory change in endogenous opioids needs to be mentioned, and the risk of causing spontaneous pain by activating Nav1.9 should also be explained.

2. Regarding statistical analysis, the sample sizes and statistical tests are now more clearly indicated. Most graphs show individual data points, and I also appreciate that the authors have provided all of the raw data. This is commendable. That said, the submission guidelines specify that F and T values should be reported, along with exact values for p, which has not been done. Results are ANOVAs (F or p values) are not reported at all; in many cases, 2-way ANOVAs (factor 1 = genotype; factor 2 = treatment) seem more appropriate than 1-way, but were not used. Most importantly, I do not think it is appropriate to not test the normality for any sample smaller than 12, and simply assume the distributions are normal and proceed with parametric tests. All of this to say that improvements are still needed.

3. The calcium imaging experiments have been removed from the paper. I appreciate the results of new experiments provided in the response.

4. Regarding the inclusion/exclusion of data for Figures 2d and g, the authors responded that they did not remove HpTx1 non-responsive cells, and well I trust that this is the case for panels a,b,c and e, why is the n so much smaller in panels d and g? Based on line 125-127 in the text, they write "in 8 out of 15 neurons with reduced rheobase...", indicating that they used one metric of increased excitability (i.e. reduced rheobase) to select neurons, for which they then used a different metric (i.e. spike count) to show that excitability is increased. This is circular reasoning. Showing that only a certain cell type (e.g. peptidergic on non-peptidergic neurons) is affected by the toxin, and then comparing the increase in spike count within each cell type would be appropriate; simply separating responsive and non-responsive cells, and assessing the change in spike count in the former is NOT appropriate.

5. The authors now cite Sanguinetti et al. on line 82. This is a big improvement, although I still think this information should be mentioned in the Introduction, when the authors write (line 65) that they "discovered" a spider peptide toxin.

6. I am unsatisfied with the interpretation that potassium channels are not affected. Sanguinetti tested Kv4.2 and Kv1.4 because those channels are expressed by cardiac myocytes; they did not rule out effects on other potassium channels. Indeed, Kv4.2 is largely absent from DRG neurons but many of those neurons express Kv4.1 and a host of other Kv channels might also be affected

by this toxin. In that respect, for extended Fig. 1b, it would have been far more relevant to test Kv4.1 channels. But my biggest problem is with extended Fig. 1c. This shows a single voltage clamp step, the details of which are not reported. Based on the details of the voltage clamp protocol, certain K channels might have been inactivated before the main pulse, in which case no effect of the toxin would be evident. Furthermore, by stepping the voltage very high, certain large currents can obscure subtle changes in smaller currents active at voltages near threshold. In short, the authors have not done the appropriate experiments to confidently say that Kv channels are not affected, and I think this is a critical aspect of their story (i.e. that all effects are via modulation of Nav channels). Their data do not convincingly show that.

7. The question about blinding has been answered.

8. I have not checked the citations, but I will take the authors word that they have improved this.

9. The answer about the liquid junction potential does not make sense. On the axopatch, the current can be zeroed before patching, but once the whole-cell configuration is achieved, a junction potential arises from the differential movement of positive and negative charges. One would normally calculate the junction potential (e.g. JPCalc) based on the pipette solution. Did the authors do this?

Additionally:

-Line 94, the authors claim that their results are consistent with Nassar et al. (PNAS 2004). That is inaccurate. Nassar et al. reported changes in both mechanical and thermal pain thresholds. A more recent paper (Shields et al. J Neurosci 2018) is also relevant here.

Reviewer #3 (Remarks to the Author):

The authors have provided important additional new information in response to the reviewers concerns. That said, concerns still remain.

1) The writing is still not consistent with proper English language usage. The additional text in the first two paragraphs contains many ideas but they are poorly organized, and not logically connected.

2) As I read the revised MS, it does not appear that the authors have adequately addressed a major concern of reviewer 1 with regards to the "premise that CIP could be treated using a toxin such as HpTx1 seems far-fetched." I defer to the original reviewer as to whether they are satisfied with the response provided.

3) With respect to the title of Table 3, by "current threshold", do the authors mean rheobase?

4) P. 6, line 120... "Please note 15 out of the 30 DRG neurons tested (50.0%) exhibited decreased rheobase, 40% were unchanged and 10% had increased rheobase in the presence of HpTx1." The header to this section is "HpTx1 activates peripheral small DRG neurons in Nav1.7-KO mice" but this is "overselling" the observation - only half of small DRG neurons demonstrate activation (i.e., have decreased rheobase) - it would be helpful to understand what makes these small DRG neurons different from the 50% that are either unaffected or show an increase in rheobase.

5) the inhibition of Kv4.2 by HpTx1 (this MS and Sanguinetti et al) is not irrelevant with respect to regulating excitability and pain plasticity (see for example Hu et al Neuron 2006); the interaction between Nav and Kv currents is not adequately addressed - for instance, is there any evidence of compensatory changes in channel expression

Reviewers' comments:

Reviewer #1 (Remarks to the Author):

the revisions have made this paper stronger. now acceptable in my view.

R: We deeply appreciate your comments, which encourage us to make a new revision according to the reviewers' comments.

Reviewer #2 (Remarks to the Author):

Most of my original concerns have not been adequately addressed by the revisions. There is a huge amount of work presented in this study but critical gaps remain. As a result, the interpretation of the data is still quite speculative, in my opinion. The numbered points correspond to the points in my original review.

R: Thank you for the comments. We are sorry we did not adequately address your concerns in our last revision. In this revised manuscript, we tried our best to address your concerns. Thank you for your time in reviewing this manuscript.

1. My first major comment about whether this toxin (or some other more specific manipulation of Nav1.9) is a viable treatment for CIP remains. The authors have added wording to specify Nav1.7-related CIP but I still think that this is a big stretch. Notably, Minett et al. (Nature Comm 2015) showed that endogenous opioids are upregulated after Nav1.7 deletion. This paper is not cited. According to it, naloxone could help restore pain perception in Nav1.7-null mice. If the author wish to emphasize that treatment of Nav1.7-related CIP, then the compensatory change in endogenous opioids needs to be mentioned, and the risk of causing spontaneous pain by activating Nav1.9 should also be explained.

R: We thank this reviewer for pointing this out again.

We agree with this reviewer's viewpoint. In this study, it is speculative or subjective to show that "this toxin is a viable treatment of CIP" without much more data regarding efficacy and safety. We removed such a statement in the discussion in the revised manuscript. We emphasize that the activation of Nav1.9 could compensate the loss of Nav1.7 function and restore the pain reaction in the Nav1.7-related CIP, which might provide an alternative to treat Nav1.7-related CIP. HpTx1 acted as a useful pharmacological probe to achieve this conclusion in this investigation.

We cited the study by Minett et al ¹. Combining our data, both studies suggest that a practical treatment strategy for Nav1.7-related CIP may be through targeting the different pain signaling pathways other than Nav1.7 itself. Minett et al. also found that the upregulation of endogenous opioids is possibly derived from lowered levels of intracellular sodium due to lacking functional Nav1.7, so it is interesting to test if enhanced Nav1.9 activity could reverse the increase of endogenous opioids in Nav1.7-related CIP. On the

other hand, we also explained the possible risks brought about by the activation of Nav1.9. In our future study, we will attempt to address these two issues. We have also added the discussion in the revised text. Please see page 18-19, lines 404 to 413 and 418 to 423.

References

1. Minett, M. S. *et al.* Endogenous opioids contribute to insensitivity to pain in humans and mice lacking sodium channel Nav1.7. *Nature communications* **6**, 8967, doi:10.1038/ncomms9967 (2015).

2.Regarding statistical analysis, the sample sizes and statistical tests are now more clearly indicated. Most graphs show individual data points, and I also appreciate that the authors have provided all of the raw data. This is commendable. That said, the submission guidelines specify that F and T values should be reported, along with exact values for p, which has not been done. Results are ANOVAs (F or p values) are not reported at all; in many cases, 2-way ANOVAs (factor 1 = genotype; factor 2 = treatment) seem more appropriate than 1-way, but were not used. Most importantly, I do not think it is appropriate to not test the normality for any sample smaller than 12, and simply assume the distributions are normal and proceed with parametric tests. All of this to say that improvements are still needed.

R: Many thanks for your professional comments. We modified statistical analyses accordingly. In detail, One-way ANOVA (Figure 1h, Figure 6c and Extended Data Figure 6) and two-way ANOVA (Figure 1c-e, g; Figure 2a-d, g; Figure 3e, h; Figure 4c-f; Figure 5a-e; Extended Data Figure 2a) were used to assess the difference between multiple groups. Two grouped data were analyzed by the Kolmogorov-Smirnov normality test before t-test analysis. If the data were normally distributed, a parametric t-test was used. Otherwise, a non-parametric t-test was used. In figure legends, the statistical method to a specific experiment was mentioned, and the F, t, df (degree of freedom) and P values were also shown. Significant levels were set at $P < 0.05$ and the exact P values are presented in Supplementary Data 1. All these modifications are mentioned in the Figure legend, the Method section, and the Reporting Summary.

3.The calcium imaging experiments have been removed from the paper. I appreciate the results of new experiments provided in the response.

R: Thanks.

4.Regarding the inclusion/exclusion of data for Figures 2d and g, the authors responded that they did not remove HpTx1 non-responsive cells, and well I trust that this is the case for panels a,b,c and e, why is the n so much smaller in panels d and g? Based on line125-127 in the text, they write “in 8 out of 15 neurons with reduced rheobase...”, indicating that they used one metric of increased excitability (i.e. reduced rheobase) to select neurons, for which they then used a different metric (i.e. spike count) to show that

excitability is increased. This is circular reasoning. Showing that only a certain cell type (e.g. peptidergic on non-peptidergic neurons) is affected by the toxin, and then comparing the increase in spike count within each cell type would be appropriate; simply separating responsive and non-responsive cells, and assessing the change in spike count in the former is NOT appropriate.

R: We thank the reviewer for pointing this out.

In the determination of rheobase and repetitive-firing of small DRG neurons with the treatment of the toxin, we referred to the methods reported by Amsalem et al. (*EMBO J*, 2018)¹ and Osteen et al. (*Nature*, 2016)². Rheobase (current threshold) is designated as the minimum current injection required to achieve the firing of a single action potential. Reduced rheobase has been considered as the increase of neuron excitability. In our experiments, the rheobases of small DRG neurons were determined by injecting current pulses from 0 pA to 140 pA with 20 pA increment. We found that approximately 50% of small DRG neurons displayed decreased rheobase in the presence of HpTx1, suggesting that HpTx1 treatment increased these neuron excitabilities. We had no direct evidence to explain the type of these neurons. However, our data showed HpTx1-induced excitability depended on the expression and activation of Nav1.9. It has been accepted that Nav1.9 is preferentially expressed in non-myelinated nonpeptidergic small DRG neurons, and in our experiments, Nav1.9 expression in DRG neurons from WT mice showed an approximately 59.6% overlap with the expression of IB4, a marker for small, unmyelinated nonpeptidergic fibres. These data suggest that these responsive neurons might be nonpeptidergic small DRG neurons expressing Nav1.9 at a high or appreciable level.

Next, we want to know if the toxin reduced the rheobase and at the same time enhanced repetitive firing. We found 10 out of 15 neurons with reduced rheobase demonstrated increased firing count in the presence of the toxin (we rechecked our data and found 10, not 8 neurons had such an effect with the treatment of toxin. We modified the revised manuscript accordingly). 5 neurons fired just a single AP regardless of the injection currents in the control group, and the toxin treatment could not induce repetitive firing. Therefore, for Figure 2d, only responsive neurons were included for analysis and the other 5 neurons were excluded because we thought this panel was related to the repetitive firing and showing repetitive data would be suitable. We apologize for our misunderstanding. In the revised manuscript, for the new Figure 2d and Figure 2g, all data for the 15 and 19 neurons were analyzed, respectively.

References

1. Amsalem, M., Poilbout, C., Ferracci, G., Delmas, P. & Padilla, F. Membrane cholesterol depletion as a trigger of Nav1.9 channel-mediated inflammatory pain. *The EMBO journal* **37**, doi:10.15252/embj.201797349 (2018).
2. Osteen, J. D. *et al.* Selective spider toxins reveal a role for the Nav1.1 channel in mechanical pain. *Nature* **534**, 494-499, doi:10.1038/nature17976 (2016).

5. The authors now cite Sanguinetti et al. on line 82. This is a big improvement, although I

still think this information should be mentioned in the Introduction, when the authors write (line 65) that they “discovered” a spider peptide toxin.

R: Thanks for this suggestion. We have mentioned this information in the Introduction as “HpTx1 was previously identified as an inhibitor of the voltage-gated potassium channel (Kv) Kv4.2³⁰, but our study showed that it is also a Nav modulator.” Please see page 3, lines 65 to 67.

6.I am unsatisfied with the interpretation that potassium channels are not affected. Sanguinetti tested Kv4.2 and Kv1.4 because those channels are expressed by cardiac myocytes; they did not rule out effects on other potassium channels. Indeed, Kv4.2 is largely absent from DRG neurons but many of those neurons express Kv4.1 and a host of other Kv channels might also be affected by this toxin. In that respect, for extended Fig. 1b, it would have been far more relevant to test Kv4.1 channels. But my biggest problem is with extended Fig. 1c. This shows a single voltage clamp step, the details of which are not reported. Based on the details of the voltage clamp protocol, certain K channels might have been inactivated before the main pulse, in which case no effect of the toxin would be evident. Furthermore, by stepping the voltage very high, certain large currents can obscure subtle changes in smaller currents active at voltages near threshold. In short, the authors have not done the appropriate experiments to confidently say that Kv channels are not affected, and I think this is a critical aspect of their story (i.e. that all effects are via modulation of Nav channels). Their data do not convincingly show that.

R: We thank the reviewer for this important comment.

We tested the effect of HpTx1 on Kv1.4 and Kv4.1. As shown in **Fig.R1**, the Kv1.4 and Kv4.1 channel currents are not significantly affected by 5 μ M HpTx1 (a saturating dose on Kv4.2). Importantly, as shown in **Extended Data Fig.1c** in the revised manuscript, we conducted the DRG voltage-gated potassium channels experiment again and counted more cells. HpTx1 could not affect voltage-gated potassium currents of mouse DRG neurons when held in -80 mV, -100 mV or -120 mV. It did not affect the voltage-current curves of DRG voltage-gated potassium channels, indicating that the toxin could not affect the voltage-gated potassium channel currents evoked at different voltages (from -70 mV to +70 mV). Furthermore, the *in vitro* and *in vivo* Nav1.9-KO experiments indicated that HpTx1 enhanced the excitability of primary afferent neurons and elicited pain most likely dependent on the expression and activation of Nav1.9, and we assumed that the voltage-gated potassium channel inhibition might not be the primary reason for pain reactions induced by HpTx1. Please see page 7, lines 146 to 152.

Figure.R1 The Kv1.4 (a) and Kv4.1 (b) currents are not significantly affected by 5 μ M HpTx1. Representative current traces (left) from HEK-293T cells expressing Kv1.4 or Kv4.1 in the absence (black) and presence of 5 μ M HpTx1 (red). Currents were elicited by depolarization to 20 mV from the holding potential of -80 mV. The current-voltage curves also (right, n=4 for Kv1.4, n=5 for Kv4.1) show no significant difference. Current amplitudes were normalized to the control peak amplitude. Currents were elicited by depolarization to various potentials ranging from -70 mV to +70 mV at increments of +10 mV from a holding potential of -80 mV.

7.The question about blinding has been answered.

R: Thanks.

8.I have not checked the citations, but I will take the authors word that they have improved this.

R: Thanks. We check the citations carefully in the revised manuscript.

9.The answer about the liquid junction potential does not make sense. On the axopatch, the current can be zeroed before patching, but once the whole-cell configuration is achieved, a junction potential arises from the differential movement of positive and negative charges. One would normally calculate the junction potential (e.g. JPCalc) based on the pipette solution. Did the authors do this?

R: Yes, we used the Henderson equation to calculate the junction potential based on the ionic strength of the bath and pipette solution. This was mentioned in the section of "Material and methods" in the revised manuscript. Please see page 22 to 23, lines 492 to 494.

Additionally:

-Line 94, the authors claim that their results are consistent with Nassar et al. (PNAS 2004). That is inaccurate. Nassar et al. reported changes in both mechanical and thermal pain thresholds. A more recent paper (Shields et al. J Neurosci 2018) is also relevant here.

R: Our mistake. Thanks for pointing it out.

We have corrected it in the revised manuscript. Our results are consistent with those in the study by Minett et al ¹. Minett et al. examined the pain behaviours in three Nav1.7 knockout mice lines [Nav1.7^{Advill} (deleting Nav1.7 in all sensory neurons), Nav1.7^{Wnt1} (deleting Nav1.7 in sensory and sympathetic neurons), and Nav1.7^{Nav1.8}]. All three Nav1.7 KO mouse strains showed pronounced analgesia in terms of noxious mechanosensation. However, thermal deficits were only found in the Nav1.7^{Advill} and Nav1.7^{Wnt1} mice, but not in Nav1.7^{Nav1.8} mice, in contrast to the data of Nassar et al ². Shields et al. ³ found that behavioral responses to most modalities of noxious stimulus are abolished following adult deletion of Nav1.7.

In this study, we used the Nav1.7^{Nav1.8} mice and determined the pain phenotypes are the same as those in the study by Minett et al ¹.

References

1. Minett, M. S. *et al.* Distinct Nav1.7-dependent pain sensations require different sets of sensory and sympathetic neurons. *Nature communications* **3**, 791, doi:10.1038/ncomms1795 (2012).
2. Nassar, M. A. *et al.* Nociceptor-specific gene deletion reveals a major role for Nav1.7 (PN1) in acute and inflammatory pain. *Proceedings of the National Academy of Sciences of the United States of America* **101**, 12706-12711, doi:10.1073/pnas.0404915101 (2004).
3. Shields et al. Insensitivity to pain upon adult-onset deletion of Nav1.7 or its blockade with selective Inhibitors. *The Journal of Neuroscience*, **38** (47), 10180 –10201, (2018).

Reviewer #3 (Remarks to the Author):

The authors have provided important additional new information in response to the reviewers concerns. That said, concerns still remain.

R: Thank you for the comments. In this revised manuscript, we hope the concerns have been addressed. Thank you for your time in reviewing our manuscript.

1) The writing is still not consistent with proper English language usage. The additional text in the first two paragraphs contains many ideas but they are poorly organized, and not logically connected.

R: The revised manuscript has been edited by Nature Research Editing Service, so we hope it can meet the journal's standard. Also, we have rewritten the Introduction section in the revised manuscript.

SPRINGER NATURE
Author Services Editing Certificate

This document certifies that the manuscript
Activating Nav1.9 Recovers Pain Responses in Nav1.7-related CIP as Revealed by a Spider Toxin
prepared by the authors
Xi Zhou, Tingbin Ma, Luyao Yang, Shuijiao Peng, Lulu Li, Zhouquan Wang, Zhen Xiao, Qingfeng Zhang, Li Wang, Yazhou Huang, Minzhi Chen,...
was edited for proper English language, grammar, punctuation, spelling, and overall style
by one or more of the highly qualified native English speaking editors at SNAS.
This certificate was issued on **February 5, 2020** and may be verified
on the SNAS website using the verification code **9327-7CF8-0B8A-893D-D02P**.

Neither the research content nor the authors' intentions were altered in any way during the editing process. Documents receiving this certification should be English-ready for publication; however, the author has the ability to accept or reject our suggestions and changes. To verify the final SNAS edited version, please visit our verification page at secure.authorservices.springernature.com/certificate/verify.
If you have any questions or concerns about this edited document, please contact SNAS at support@as.springernature.com.

SNAS provides a range of editing, translation, and manuscript services for researchers and publishers around the world.
For more information about our company, services, and partner discounts, please visit authorservices.springernature.com.

2) As I read the revised MS, it does not appear that the authors have adequately addressed a major concern of reviewer 1 with regards to the "premise that CIP could be treated using a toxin such as HPtx1 seems far-fetched." I defer to the original reviewer as to whether they are satisfied with the response provided.

R: We thank the reviewer for pointing this out.

We agree with this reviewer's viewpoint. In this study, it is speculative or subjective to assert that "CIP could be treated using a toxin such as HpTx1" without much more data regarding efficacy and safety. Therefore, we removed such a statement in the discussion

in the revised manuscript. We only emphasize that the activation of Nav1.9 could compensate for the loss of Nav1.7 function and restore the pain reaction in the Nav1.7-related CIP, which might provide an alternative to treat Nav1.7-related CIP. HpTx1 acted as a useful pharmacological probe to achieve this conclusion in this investigation.

According to the second reviewer's advice, we cited the study by Minett et al.¹. Combining our data, both studies suggest that a practical treatment strategy for Nav1.7-related CIP may be through targeting the different pain signaling pathways other than Nav1.7 itself. Minett et al. also found that the upregulation of endogenous opioids is possibly derived from lowered levels of intracellular sodium due to lacking functional Nav1.7, so it is interesting to test if enhanced Nav1.9 activity could reverse the increase of endogenous opioids in Nav1.7-related CIP. On the other hand, we also explained the possible risks brought about by the activation of Nav1.9. In our future study, we will attempt to address these two issues. We have also added the discussion in the revised text. Please see page 18-19, lines 404 to 413 and 418-423.

References

1. Minett, M. S. *et al.* Endogenous opioids contribute to insensitivity to pain in humans and mice lacking sodium channel Nav1.7. *Nature communications* **6**, 8967, doi:10.1038/ncomms9967 (2015).

3) With respect to the title of Table 3, by "current threshold", do the authors mean rheobase?

R: Yes, it is rheobase. We have modified this in the revised manuscript.

4) P. 6, line 120... "Please note 15 out of the 30 DRG neurons tested (50.0%) exhibited decreased rheobase, 40% were unchanged and 10% had increased rheobase in the presence of HpTx1." The header to this section is "HpTx1 activates peripheral small DRG neurons in Nav1.7-KO mice" but this is "overselling" the observation - only half of small DRG neurons demonstrate activation (i.e., have decreased rheobase) - it would be helpful to understand what makes these small DRG neurons different from the 50% that are either unaffected or show an increase in rheobase.

R: We thank the reviewer for pointing this out.

First, we changed the header of the section into "HpTx1 activates peripheral some small DRG neurons in Nav1.7-KO mice".

Second, we proposed that 50.0% neurons with decreased rheobase might be non-myelinated nonpeptidergic small-diameter DRG neurons expressing Nav1.9, while the other 50% neurons might be low or non-Nav1.9 expressed neurons (we assumed that among these neurons, some might express Nav1.7 and therefore display increased rheobase because of Nav1.7 inhibition by the toxin, and some might have low or no Nav1.7 expression and therefore be unaffected by the toxin). This assumption is derived from *in vitro* and *in vivo* experiments, in which Nav1.9 has been proved to be required for HpTx1 enhancing neurons excitability and inducing pain. Not all DRG neurons express Nav1.9 at high or appreciable levels. Nav1.9 is preferentially expressed in non-myelinated

nonpeptidergic small-diameter DRG neurons¹, which was also confirmed in our study, showing IB4 and Nav1.9 had co-expression and shared 59% overlap. Also, HpTx1 failed to produce neurogenic inflammation in the injected hind paws. This effect is like that of Hm1a which is a Nav1.1 selective modulator from spider venom and elicits robust pain behaviours without neurogenic inflammation². The authors suggest that Hm1a elicits pain by activating nonpeptidergic neurons.

In the second paragraph in Page 11, line 239 to 241, we made some refinements briefly.

References

1. Salvatierra, J. *et al.* A disease mutation reveals a role for Nav1.9 in acute itch. *The Journal of clinical investigation* **128**, 5434-5447, doi:10.1172/JCI122481 (2018).
2. Osteen, J. D. *et al.* Selective spider toxins reveal a role for the Nav1.1 channel in mechanical pain. *Nature* **534**, 494-499, doi:10.1038/nature17976 (2016).

5) the inhibition of Kv4.2 by HpTx1 (this MS and Sanguinetti et al) is not irrelevant with respect to regulating excitability and pain plasticity (see for example Hu et al Neuron 2006); the interaction between Nav and Kv currents is not adequately addressed - for instance, is there any evidence of compensatory changes in channel expression

R: We thank the reviewer for this suggestion.

Kv4.2 is low expressed in the peripheral nervous systems¹ and our data showed that HpTx1 has no inhibition on voltage-gated Kv channels in mouse DRG neurons. Therefore, the Kv inhibitory activity of HpTx1 was not considered the main point in our study. We just focused on the three sodium channels and their interactions. (Please see page 7, lines 146 to 152.)

The interaction between Kvs and Navs in pain signaling is very interesting since it will provide new insight into pain signaling and drug development. Considering HpTx1 has inhibition on Kv4.2 and Nav1.7 and activation on Nav1.9, using HpTx1 as a probe would also be helpful for the elucidation of pain signal transduction and drug development. We expect to perform such a study in the future, which will be an interesting study as you mentioned.

References

1. Phuket, T. R. & Covarrubias, M. Kv4 Channels Underlie the Subthreshold-Operating A-type K-current in Nociceptive Dorsal Root Ganglion Neurons. *Frontiers in molecular neuroscience* **2, 3**, doi:10.3389/neuro.02.003.2009 (2009).

REVIEWERS' COMMENTS:

Reviewer #1 (Remarks to the Author):

This paper will be of interest to a broad audience. In my view it is ready for publication

Reviewer #3 (Remarks to the Author):

Prior issues appear to have been addressed.

There are several minor points that still need to be addressed.

1. Abstract - lines 24-27. That HpTx1 was first identified as an inhibitor of Kv4.2 needs to be included here, otherwise the emphasis on NaV channels is disingenuous.
2. p. 7, lines 149-150 "in periphery neurons" should be changed to "in peripheral sensory neurons"
3. Fig. 1B - what is the meaning/significance of the color coding scheme in the sequence alignment? There is no obvious pattern that I can discern; if there is meaning, then the code needs to be provided in the legend.

REVIEWERS' COMMENTS:

Reviewer #1 (Remarks to the Author):

This paper will be of interest to a broad audience. In my view it is ready for publication

R: We deeply appreciate your comments.

Reviewer #3 (Remarks to the Author):

Prior issues appear to have been addressed.

R: We greatly appreciate the reviewer for the positive comment.

There are several minor points that still need to be addressed.

1. Abstract - lines 24-27. That HpTx1 was first identified as an inhibitor of Kv4.2 needs to be included here, otherwise the emphasis on NaV channels is disingenuous.

R: Thanks! The abstract have edited by the editor.

2. p. 7, lines 149-150 "in periphery neurons" should be changed to "in peripheral sensory neurons"

R: Thanks! We have corrected.

3. Fig. 1B - what is the meaning/significance of the color coding scheme in the sequence alignment? There is no obvious pattern that I can discern; if there is meaning, then the code needs to be provided in the legend.

R: Thanks for this suggestion. We have modified the panel in the revised manuscript.